# Genes That Associated with Action of ACTH-like Peptides with Neuroprotective Potential in Rat Brain Regions with Different Degrees of Ischemic Damage

**DOI:** 10.3390/ijms26136256

**Published:** 2025-06-28

**Authors:** Ivan B. Filippenkov, Yana Yu. Shpetko, Daria A. Ales, Vasily V. Stavchansky, Alina E. Denisova, Vadim V. Yuzhakov, Natalia K. Fomina, Leonid V. Gubsky, Lyudmila A. Andreeva, Nikolay F. Myasoedov, Svetlana A. Limborska, Lyudmila V. Dergunova

**Affiliations:** 1Laboratory of Human Molecular Genetics, National Research Center “Kurchatov Institute”, Kurchatov Sq. 2, Moscow 123182, Russia; yana.sch2014@yandex.ru (Y.Y.S.); alles_d@mail.ru (D.A.A.); bacbac@yandex.ru (V.V.S.); andr-la.img@yandex.ru (L.A.A.); myasoedov-nf.img@yandex.ru (N.F.M.); limbor.img@yandex.ru (S.A.L.); dergunova-lv.img@yandex.ru (L.V.D.); 2Department of Neurology, Neurosurgery and Medical Genetics, Pirogov Russian National Research Medical University, Ostrovitianov Str. 1, Moscow 117997, Russia; dalina543@gmail.com (A.E.D.);; 3A. Tsyb Medical Radiological Research Center, Branch of the National Medical Research Radiological Center, Ministry of Health of the Russian Federation, Koroleva Str. 4B, Obninsk 249036, Russia; yuzhvad.mail@yandex.ru (V.V.Y.); nkfomina@rambler.ru (N.K.F.); 4Federal Center for the Brain and Neurotechnologies, Federal Biomedical Agency, Ostrovitianov Str. 1, Building 10, Moscow 117997, Russia

**Keywords:** ischemic stroke, ACTH-like peptides, frontal cortex, ipsilateral striatum, tMCAO, RNA-Seq, gene network, gene expression

## Abstract

In the treatment of ischemic stroke, an innovative approach is the use of neuroprotective compounds. Natural peptides, including adrenocorticotropic hormone (ACTH), can serve as the basis for such drugs. Previously, a significant effect of non-hormonal ACTH(4-7)PGP (Semax) and ACTH(6-9)PGP peptides on the functions of the nervous system was shown. Also, while using RNA-Seq, we firstly revealed differentially expressed genes (DEGs) that associated with peptides in the penumbra-associated region of the frontal cortex (FC) of rats at 24 h after transient middle cerebral artery occlusion (tMCAO) model. Peptides significantly reduced profile disturbances caused by ischemia for almost two-thousand DEGs in FC related to the neurotransmitter and inflammatory response. Here, we studied how peptides affected the expression of genes in the striatum with an ischemic focus, predominantly. The same animals from which we previously acquired FC were used to collect striatum samples. Peptides generated fewer DEGs in the striatum than in the FC. Both peptides tended to normalize the profile of disturbances caused by ischemia for hundreds of DEGs, whereas 152 genes showed an even more affected profile in the striatum under ACTH(6-9)PGP action. These DEGs were associated with inflammation, predominantly. About hundred genes were overlapped between both peptides in both tissues and were associated with neuroactive ligand-receptor interaction, predominantly. Thus, genes that are associated with the ACTH-like peptide action in rat brain regions with varying levels of ischemia injury were identified. Moreover, differential spatial regulation of the ischemia process in the rat brain at the transcriptome levels was discovered under peptides with different ACTH structures. We suppose that our results may be useful for selecting more effective neuroprotective drug structures in accordance with their specific tissue/damage therapeutic impact.

## 1. Introduction

The multifactorial disease known as ischemic stroke causes significant neurological impairment and damage to brain tissue. Strokes affect tens of millions of people annually [1,2]. In addition to classical approaches to medical care (thrombolysis, reperfusion, anticoagulation), much attention is paid to the development of neuroprotective agents after ischemic stroke [3,4,5,6].

It is known that the neuroendocrine system response occurs during ischemic stroke events and plays a decisive role in vulnerability and susceptibility to stroke [7,8,9]. Several peptide hormones, in addition to their hormonal function, have the ability to provide a neuroprotective effect after stroke. These include melatonin, orexin-A, ghrelin, hepcidin, α-melanocyte-stimulating hormone (α-MSH), and others [10,11,12,13,14,15,16]. Thus, a synthetic peptide based on α-MSH (TAT-HSA-α-MSH) has gained popularity [17]. Our team is also actively searching for peptide-based drugs and studying their properties. We have previously studied the spectrum of neuroprotective activity of a synthetic peptide Semax (Met-Glu-His-Phe-Pro-Gly-Pro), including its transcriptomic effect in ischemic brain cells [18]. Semax is based on adrenocorticotropic hormone (ACTH). It has an ACTH(4–7) fragment, and the tripeptide Pro-Gly-Pro (PGP) is added to stabilize the C-terminus. The peptide was developed as a nootropic and neuroprotective agent without hormonal activity and other side effects. Prior research has shown that Semax improves animal survival after ischemic stroke and reduces the degree of neurological deficit [19]. By studying its effect on the transcriptome of ischemic brain cells, we found that Semax initiated a neurotransmitter and anti-inflammatory response and compensated for the impairments caused by ischemia. We identified dozens of signaling pathways associated with DEGs influenced by Semax in ischemia [18]. The list of signaling pathways includes calcium, dopaminergic and cholinergic synapses, as well as phagosome, cytokine, neutrophil degranulation, and other pathways [18]. It is currently believed that peptides, including Semax, are able to provide allosteric interactions of the patterns of peptide metabolites (synactone) with various types of receptors [20]. ACTH(6-9)PGP (His-Phe-Arg-Trp-Pro-Gly-Pro), another synthetic adrenocorticotropic-like peptide, recently demonstrated nootropic action [21]. A neuroprotective effect of ACTH(6-9)PGP was discovered using animal studies [21].

Our studies revealed that Semax and ACTH(6-9)PGP had some interesting impact on the rat brain under conditions of the transient middle cerebral artery occlusion (tMCAO) model. Therefore, at 4.5 h and primarily 24 h following tMCAO, peptides were associated with a correction of the gene-expression patterns that are disturbed during cerebral ischemia in the penumbra-associated region of the frontal cortex (FC) [22,23]. Peptide-related pattern genes were categorized into a neurotransmitter cluster (NC) and an inflammatory cluster (IC). And we called this effect of the peptides on gene expression during ischemia a compensatory effect. Furthermore, we identified that peptides cause neuroglial cell proliferation and vascularization of brain tissues in perifocal zones at 24 h after tMCAO in a histological study [24].

Here, using RNA-Seq we aimed to study the effect of Semax and ACTH(6-9)PGP peptides on gene expression in the ischemia focus-containing striatum. The same animals from which we previously acquired FC were used to collect striatum samples [23]. Thus, at 24 h after tMCAO, we compared the differentially expressed gene (DEG) profiles of two rat brain regions. Histological examination (HE) and magnetic resonance imaging (MRI) data were used to select the tissues. In consequence, numerous DEGs were identified both under ischemia conditions, as well as after peptide administration at 24 h after tMCAO in striatum of rats. However, fewer DEGs were detected in the striatum than in the FC under the influence of peptides. Furthermore, if peptides significantly reduced profile disturbances caused by ischemia of more than a thousand DEGs in the FC, then less than three-hundred DEGs were compensated by peptides in ischemic striatum. Additionally, under ACTH(6-9)PGP action, there were 152 genes that showed an even more affected profile in the striatum than was induced by ischemia itself. These DEGs were predominantly associated with inflammation. About one-hundred genes were overlapped between the action of both peptides in both tissues and were associated with neuroactive ligand–receptor interaction, predominantly. Consequently, genes associated with the action of ACTH-like peptides were identified in the rat brain areas with varying degrees of ischemia injury. Moreover, differential spatial regulation of the ischemic process in the rat brain at the transcriptome level was revealed under peptides with different ACTH structures. We suppose that our results may be useful for selecting more effective neuroprotective drug structures in accordance with their specific tissue/damage therapeutic impact.

## 2. Results

### 2.1. Histological Examination of Rat Brain Samples

The ipsilateral striatum’s histological examination (HE) using Nissl staining is displayed in Figure 1a. At 24 h following tMCAO, rats of the IR groups had ischemia damage sites that were located in the brains of the ipsilateral hemisphere (subcortex plus cortex). The striatum had a partly penumbra zone and necrotic areas (Figure 1a).

### 2.2. RNA-Seq Analysis of the Effect of Semax and ACTH(6-9)PGP Peptide on the Striatum of Rats at 24 h After tMCAO

By using RNA-Seq, we examined the transcriptional activity of the mRNAs for 17,367 genes in the rat striatum 24 h following tMCAO. In pairwise comparison of the RNA-Seq data for “ischemia-reperfusion, striatum samples” (IR-s) versus “ischemia-reperfusion + Semax, striatum samples” (IS-s) (IS-s vs. IR-s pairwise comparison), we discovered notable changes at the mRNA level for 343 differentially expressed genes (DEGs) with 144 up- and 199 downregulated genes (Appendix A). The volcano plot in Figure 1b demonstrates the variations in mRNA expression levels between the IS-s and IR-s comparison groups. We noticed that the top five most highly upregulated genes in response to IR included *Ercc2*, *Tbr1*, *Satb2*, *Neurod2*, and *Prss12*. At the same time, the top five downregulated DEGs were *Hcrtr1*, *Wnt2b*, *Gldn*, *Prok2*, and *Pkhd1l1* (Appendix A).

Additionally, in the striatum of rats subjected to IR and ACTH(6-9)PGP administration (IA-s group), we identified 735 DEGs (281 up- and 454 downregulated) compared to the saline-treated IR group (IA-s vs. IR-s pairwise comparison) (Figure 1c). The top five upregulated DEGs in IA-s vs. IR-s pairwise comparison were *Ercc2*, *Avp*, *Fam71a*, *Slpi*, and *Mmp3*, whereas the top five most considerably downregulated genes were *Scn5a*, *Prok2*, *Gp5*, *Hcrtr1*, and *Pkhd1l1* (Appendix A). Thus, we discovered that the IR-related gene-expression profile was modulated by Semax and ACTH(6-9)PGP peptides.

We observed that 241 DEGs overlapped in the IS-s vs. IR-s and IA-s vs. IR-s pairwise comparisons (Figure 1d, Appendix A). Venn diagrams (Figure 1e,f) are used for comparison between only upregulated genes and only downregulated genes under both conditions. Only *Cck* gene-encoding cholecystokinin oppositely directly changed their mRNA level IS-s vs. IR-s and IA-s vs. IR-s, whereas all other genes co-directly (in the same direction) changed their mRNA level in both cases. The top 10 overlapping genes with the largest fold change in IA-s vs. IR-s included up- (*Ercc2*, *Fam71a*, *Coch*, *Map2k6*, and *Dbp*) and down- (*Tll1*, *Prok2*, *Gp5*, *Hcrtr1*, and *Pkhd1l1*) regulated DEGs in both peptide actions.

In addition, the Venn diagram (Figure 1c) displays two relative complements for IS-s vs. IR-s and IA-s vs. IR-s pairwise comparisons. So, we found 102 DEGs that altered the expression levels only under Semax action at 24 h after tMCAO in the striatum. The top 10 genes from this list that had the greatest fold change in IS-s vs. IR-s included *Tbr1*, *Slc17a7*, *Hes5*, *Sulf1*, *Nr4a2*, as up- and *Asb2*, *Tagln*, *Rem2*, *Il1r2*, *Egr4* as downregulated genes in IR-s vs. “sham operation, striatum samples” (SO-s). Additionally, there were DEGs that altered the expression levels only after ACTH(6-9)PGP administration versus saline at 24 h after tMCAO. The top 10 genes from this list that had the greatest fold change in IA-s vs. IR-s included *Avp*, *Slpi*, *Cybb*, *Oxt*, *S100a9* as up- and *Zic4*, *Trpc4*, *Hs3st2*, *Crhr2*, *Scn5a* as downregulated genes in IA-s vs. IR-s. Full lists of DEGs unique to the IS-s vs. IR-s and IA-s vs. IR-s pairwise comparisons are shown in Appendix A, respectively.

### 2.3. Differences in Rat Brain Transcriptomes Following Ischemia and After Peptide Administration in Striatum and FC at 24 h After tMCAO

Previously, using RNA-Seq, we identified 4409 DEGs (2364 up- and 2045 downregulated) under IR conditions at 24 h after tMCAO versus sham operation (IR-s vs. SO-s) in striatum [25]. Here, we revealed common genes between peptides and IR action. So, under Semax action, there were 269 overlapping DEGs in the IS-s vs. IR-s and IR-s vs. SO-s groups (Figure 2a, Appendix A). A comparison between only upregulated genes and only downregulated genes under both conditions showed only 8 upregulated DEGs (*Fos*, *Aqp5*, *Dusp10*, *Irf8*, *Btg2*, *Col9a1*, *Edn1* and *Zbtb7b*) and one downregulated DEG encoded double C-2-like domain beta (*Doc2b*). In summary, these 9 genes co-directly changed their mRNA level in both instances (Figure 2d). Hence, 260 out of 269 overlapping DEGs (e.g., *Ercc2*, *Tbr1*, *Neurod2*, *Prss12*, *Slc17a7*, *Gp5*, *Hcrtr1*, *Wnt2b*, *Gldn*, *Prok2*) oppositely directly changed their expression level in IS-s vs. IR-s and IR-s vs. SO-s pairwise comparisons (Figure 2d). Thus, Semax tended to normalize disturbances of gene-expression profiles predominantly caused by ischemia.

Additionally, under ACTH(6-9)PGP action, we revealed 446 overlapping DEGs in the IA-s vs. IR-s and IR-s vs. SO-s groups (Figure 2b, Appendix A). Again, among overlapping DEGs, there were 294 DEGs that oppositely directly (in the opposite directions) changed their mRNA level in the IA-s vs. IR-s and IR-s vs. SO-s pairwise comparisons (Figure 2d). As can be seen, the genes in which expression was increased by IR were downregulated by peptide and included 182 genes (e.g., *Crhr2*, *Tll1*, *Prok2*, *Gp5*, *Hcrtr1*), and the genes in which expression was decreased by IR were upregulated by peptide and included 112 genes (e.g., *Ercc2*, *Slc22a13*, *Coch*, *Map2k6*, *Adamtsl5*) (Appendix A).

Then, we found 152 DEGs that co-directly changed their mRNA level in the IA-s vs. IR-s and IR-s vs. SO-s pairwise comparisons (Figure 2d). Among them, 96 DEGs co-directly upregulated their mRNA level in the IA-s vs. IR-s and IR-s vs. SO-s pairwise comparisons (Appendix A). The top 5 of them were *Avp*, *Cybb*, *S100a9*, *Thbs4*, and *Rsad2* genes. Additionally, there were 56 DEGs that co-directly downregulated their mRNA level in both cases, too (Appendix A). The top 5 of them were *Nog*, *Nppa*, *Chrna4*, *Adra2a*, and *Doc2b* genes. Thus, there were 152 out of 446 overlapping DEGs that showed an even more affected profile after ACTH(6-9)PGP treatment than was caused by IR itself in the striatum.

In addition, the triple intersection between IR-s vs. SO-s, IS-s vs. IR-s, and IA-s vs. IR-s pairwise comparisons gave 182 DEGs (Figure 2c). Most of them (173) showed compensatory action to IR by any peptides at the transcriptome level in striatum (Appendix A). They (e.g., *Gpr6*, *P2ry1*, *Map2k6*, *Adra2c*, *Fos*, *Grik2*, *Kcnh4*, *Gabrd*, *Cxcr4*, *Cd24*) can highlight common molecular bases of neuroprotective effects of studied ACTH-like peptides. Furthermore, there were 87 genes that overlapped between IR-s vs. SO-s, IS-s vs. IR-s but not IA-s vs. IR-s pairwise comparison (Figure 2c). They (e.g., *Egr2*, *Klf10*, *Ccl2*, *Htr1b*, *Adcy1*, *Cacna1h*, *Hes5*, *Chrm1*, *Camk2a*, *Neurod2*) characterized the specific effect of Semax in ischemic striatum that was also compensatory versus IR action (Appendix A). Concomitantly, 264 genes overlapped between IR-s vs. SO-s, IA-s vs. IR-s but not IS-s vs. IR-s pairwise comparisons (Figure 2c). They characterized the specific effect of ACTH(6-9)PGP peptide in striatum. This effect, identified among other things, was not only associated with compensation of the gene-expression profile impaired by ischemia but also with its enhancement of ischemia’s impact on the gene-expression pattern (e.g., *S100a9*, *Gdnf*, *Bcl2a1*, *Tlr4*, *Ccr5*, *Grin2d*, *Kcns2*, *Gabra5*, *Adra2a*, *Chrna4*) in the striatum (Appendix A).

### 2.4. Comparison of RNA-Seq Results of Semax and ACTH(6-9)PGP Action in Striatum and FC at 24 h After tMCAO

In a previous study, we used RNA-Seq to discover thousands of DEGs in FC (cut off more than 1.5, and *Padj* < 0.05) under both Semax and ACTH(6-9)PGP peptide action at 24 h following tMCAO [23]. We compared DEG sets between striatum and FC tissues for Semax (Figure 3a) and ACTH(6-9)PGP (Figure 3b) action. Venn diagrams clearly evidence that peptides generated fewer DEGs in the striatum than in the FC. Moreover, the majority of genes were unique for the action of the peptide on each of their tissues. We found only some dozens of overlapping DEGs that were modulated by peptides in both ischemic FC and striatum. Full lists of such DEGs for Semax and ACTH(6-9)PGP are shown in Appendix A, respectively. At 24 h after tMCAO, overlapping genes predominantly co-directly changed their mRNA level in both tissues. Interestingly, there were 8 DEGs, including *Fos*, *Dusp10*, *Cbln1*, *Sulf1*, *Tnfrsf11b*, *Prkar2b*, *Asb2*, and *Rsph10b* that oppositely directly changed expression in FC and striatum at 24 h after tMCAO under Semax action (Appendix A). Also, there were 46 DEGs, including inflammatory (*Ccr1*, *C1ql3*, *Tnfrsf11b*), transcription factors (*Atf3*, *Jun*), neurosignaling (*Chrm3*, *Slc25a18*, *Gpr165*), and other related genes, that oppositely directly changed expression in FC and striatum at 24 h after tMCAO under ACTH(6-9)PGP administration (Appendix A). At the same time, the intersection of 46 and 8 genes that oppositely directly changed expression in different tissues under ACTH(6-9)PGP and Semax action, respectively, amounted to only 3 genes (*Dusp10*, *Tnfrsf11b*, *Prkar2b*) (Appendix A). Thus, the action of Semax and ACTH(6-9)PGP on genes that oppositely directly changed expression in different tissues almost do not overlap. Appendix A presents a comparison of DEGs between IS-s vs. IR-s, IA-s vs. IR-s, IS-f vs. IR-f, and IA-f vs. IR-f groups. Furthermore, 111 DEGs (e.g., *Map2k6*, *Gpr6*, *Cxcl12*, *Dusp10*, *Gabrd*, *Grik2*, *Tnfrsf11b*, *Mmp9*, *Vgf*, *Hcrtr1*) overlapped between both peptides in both tissues. All of them co-directly changed expression under both peptides regardless of tissue (Appendix A) and were predominantly associated with neuroactive ligand-receptor interaction.

The hierarchical cluster analysis of all DEGs in the IR-s vs. SO-s, IS-s vs. IR-s, IA-s vs. IR-s, as well as IR-f vs. SO-f, IS-f vs. IR-f, and IA-f vs. IR-f (the index “f” denotes the frontal cortex) is illustrated in Figure 3c. The IR-induced differential gene-expression profile was mostly similar in FC and striatum at 24 h after tMCAO. At the same time, the action of peptides was noticeably different for both studied tissues. It should be noted that the peptides significantly reduced gene expression activated by IR but increased expression reduced by IR in FC. Concomitantly, in striatum, such a compensatory effect was more rarely observed and there were cases when the peptides (especially ACTH(6-9)PGP) maintained the profile disturbances initially caused by the ischemia itself.

Real-time RT-PCR (reverse transcription polymerase chain reaction) was used to study the expression patterns of *Hspb1*, *Cxcl16,* and *Casp3* genes under IR and peptide action in striatum (Figure 3d–f) and FC (Figure 3g–i), respectively. In an extended sample of animals, RT-PCR results adequately confirmed the RNA-Seq data. Namely, under IR conditions, studied genes were upregulated both in FC and striatum regions of the ipsilateral hemisphere. Concomitantly, both Semax and ACTH(6-9)PGP significantly decreased their expression level only in FC (Figure 3d–f), whereas expression changes of genes in striatum were insignificant versus saline under IR conditions (Figure 3g–i).

### 2.5. Functional Annotations of DEGs Altered in Different Comparison Groups

Using DAVID v.2021, a KEGG pathway-enrichment analysis of DEGs in IS-s vs. IR-s and IA-s vs. IR-s pairwise comparisons was conducted. Consequently, 13 and 10 KEGG pathways were identified for DEGs (*Padj* < 0.05) of IS-s vs. IR-s and IA-s vs. IR-s, respectively (Appendix A). The top five pathways that had the most significant *Padj* value in IS-s vs. IR-s were neuroactive ligand-receptor interaction, ECM-receptor interaction, calcium, circadian entrainment, and fluid shear stress and atherosclerosis (Figure 4a). The top five pathways in IA-s vs. IR-s were neuroactive ligand–receptor interaction, calcium, lipid and atherosclerosis, morphine addiction, and osteoclast differentiation (Figure 4b). There were three overlapped pathways (cAMP, neuroactive ligand-receptor interaction and calcium signaling pathway) between Semax and ACTH(6-9)PGP action (Figure 4c).

In a previous study, an analogous functional analysis was performed for DEGs in IR-s vs. SO-s of the striatum, when we found 151 KEGG pathways [25]. Furthermore, 134, 57, and 84 KEGG pathways were identified in IR-f vs. SO-f, IS-f vs. IR-f, and IA-f vs. IR-f pairwise comparisons, respectively, in FC [23]. The significant (*Padj* < 0.05) pathways identified for all comparisons are presented in Appendix A. Venn diagrams show a comparison of pathway sets between four IR-s vs. SO-s, IS-s vs. IR-s, IR-f vs. SO-f, and IS-f vs. IR-f pairwise comparisons (Appendix A), as well as between four IR-s vs. SO-s, IA-s vs. IR-s, IR-f vs. SO-f, and IA-f vs. IR-f pairwise comparisons (Appendix A). As a result, more than one hundred pathways overlapped between IR-f vs. SO-f and IR-s vs. SO-s under IR conditions. Additionally, dozens of pathways overlapped between peptides and IR action in FC. However, only 8 pathways overlapped between the IS-f vs. IR-f and IS-s vs. IR-s pairwise comparisons after Semax administration (Appendix A) and 5 pathways overlapped between the IA-f vs. IR-f and IA-s vs. IR-s pairwise comparisons after ACTH(6-9)PGP administration (Appendix A). All of them included cAMP, neuroactive ligand-receptor interaction and calcium signaling pathway. These 3 pathways also overlapped between IS-s vs. IR-s and IA-s vs. IS-s (Figure 4c). This pathway group was denominated Pathway Cluster 1 (PC1). All of them related to the neurotransmitter response. Interestingly, DEGs that were associated with these pathways were downregulated under IR regardless of being FC or striatum tissue. But only Semax tended to upregulate pathway-related DEGs in both tissues, whereas ACTH(6-9)PGP did the same only in FC (Figure 4d).

Remarkably, there were identified 10 and 7 pathways that lie in relative complement for Semax and ACTH(6-9)PGP action, respectively (Figure 4c). They were named PC2 and PC3, respectively. The Semax group included neurotransmitter cluster (NC) pathways, namely nicotine, morphine, GABAergic synapse, and inflammatory cluster (IC) pathways, namely lipid and atherosclerosis, osteoclast differentiation, yersinia infection, and malaria. As can be seen, ischemia action in both brain tissues in IR-f vs. SO-f and IR-s vs. SO-s pairwise comparisons were associated with downregulated genes of NC and upregulated genes of IC (Figure 4e). Semax predominantly compensates for IR-related gene expression disturbances in both tissues for both IC and NC (Figure 4e). The ACTH(6-9)PGP group also included NC (e.g., Axon guidance, Rap1) and IC (e.g., Chemokine, PI3K-Akt, Focal adhesion, TNF) pathways (Figure 4f). Interestingly, ACTH(6-9)PGP action magnified the effects of ischemia in the striatum and partially prevented it in the FC. So, predominantly downregulated genes were associated with NC (Figure 4f), whereas upregulated genes were associated with IC (Figure 4f) under ACTH(6-9)PGP in the striatum.

At the next stage, PC2 was considered. We calculated 289 up- and downregulated genes that originate from IA-s vs. IR-s-relative complement for IR-s vs. SO-s and IA-s vs. IR-s pairwise comparison (Figure 2b, Appendix A). Additionally, we calculated 446 up- and downregulated genes that originate from 294 overlapping genes (Appendix A) that oppositely directly (Figure 2d) and 152 genes that co-directly (Figure 2d) changed their mRNA level. As a result, we found that 294 NC genes were predominantly upregulated (Appendix A), whereas genes of IC from IA-s vs. IR-s-relative complement were partially downregulated (Appendix A). Thus, the peptide was also able to modulate IC and NC genetic response at 24 h after tMCAO and partially compensate for gene-expression disturbances in the damaged striatum. Furthermore, using DAVID, we independently analyzed 152 DEGs that co-directly changed expression by the peptide and ischemia. They were significantly associated with 8 signaling pathways (Appendix A). Such pathways, including leishmaniasis, lipid and atherosclerosis, osteoclast differentiation, toll-like receptor, and other signaling pathways, were mainly associated with IC. The ACTH(6-9)PGP increased the expression of the corresponding genes in the striatum at 24 h after ischemia (Appendix A).

Thus, both peptides tended to have compensatory effect on the gene-expression activity of IC and NC in both FC and striatum, but the ACTH(6-9)PGP peptide additionally generated some trigger action to IR through the IC genomic response of the striatum cells at 24 h after tMCAO.

### 2.6. The Involvement of the DEGs of NC and IC Associated with Effects of ACTH(6-9)PGP Peptides a Day After IR Conditions in Striatum

Our analysis showed that the effect of Semax in the striatum was less pronounced than in the FC, but in general it had a compensatory nature. Gene expression of IC was decreased, but gene expression of NC increased by Semax in contrast to IR impact. At the same time, the effect of ACTH(6-9)PGP was more heterogeneous. In the FC, its effect was compensatory, but in the striatum it showed features of an increase in the effect of IR on gene expression. We proceed to analyze genes that were associated with ACTH(6-9)PGP effects in striatum but not in FC. Furthermore, proteins of these genes were involved in ten identified pathways of IC and NC in the ischemic striatum after ACTH(6-9)PGP administration (PC2). Thus, we revealed 70 DEGs in IA-s vs. IR-s but not IA-f vs. IR-f at 24 h after tMCAO (Appendix A). Among them, there were 34 up- and 36 downregulated DEGs in IA-s vs. IR-s. The network of genes and the pathways involving their protein products are illustrated in Figure 5. In summary, the network included 70 genes, 6 NC pathways, and 4 IC pathways. The scheme shows pathway clusters (NC and IC) grouped in white ellipses. The genes are represented by rectangular blocks colored according to their differential expression in IA-s vs. IR-s pairwise comparison. The lines connecting the pathways and genes indicate the participation of the protein products of genes in the pathway functioning. Using DAVID v.2021, the gene connections for KEGG pathways from each NC and IC were re-examined. In addition, genes are grouped in accordance with their association with pathway clusters, as well as up/downregulation of their expression in IA-s vs. IR-s. It should be noted that we revealed four gene groups in the network using Cytoscape 3.9.2 (Figure 5, Appendix A). The first group included 24 upregulated DEGs that were associated with IC. The second group included 33 downregulated DEGs that were associated with NC. The third group included 10 upregulated DEGs that were associated with NC. Finally, the fourth group included 3 downregulated DEGs that were associated with IC. In summary, downregulated DEGs were predominantly associated with NC, whereas upregulated DEGs were predominantly associated with IC in IA-s vs. IR-s. However, the ACTH(6-9)PGP peptide also initiated a compensatory effect on downregulated IC and upregulated NC genes, reversing ischemia impact (third and fourth gene groups). Each group of genes included those that had the maximum number of connections with signaling pathways within the cluster. So, in the first group there were *Fos*, *Tlr4*, and *Pik3cd* genes, which had 3 connections each with IC. In the second group there were *Cacna1a*, *Grin2d*, *Gabbr2*, *Gabra3*, *Gabra5* genes, which had 4 connections each with NC. In the third group there were *Adora2a* and *Prkcg* genes (3 connections with NC). In the fourth group there were *Mapk11* and *Nfatc2* genes (3 connections with IC). In addition, there were *Camk2d*, *Fos*, *Pik3cd*, *Vav3,* and *Calcr* genes that were associated with pathways from different clusters NC and IC. Thus, the network (Figure 5, Appendix A) reflected the striatum-specific features of the involvement of genes that were associated with ACTH(6-9)PGP effects in the formation of inflammatory and neurosignaling responses a day after IR conditions in rat damaged striatum.

## 3. Discussion

In this study, we analyzed the effect of the ACTH-like peptides on the transcriptome of rat brain cells at 24 h after tMCAO. We previously showed that the Semax and ACTH(6-9)PGP peptides alter the expression of about 2000 genes in the frontal cortex (FC) containing penumbra and viable cells [23]. Most of these genes overlapped with those that changed expression in response to ischemia in this brain region. Moreover, those that reduced expression during IR increased it under the action of peptides and vice versa, those that increased expression during IR decreased it under the action of peptides in FC [23]. According to MRI and HE data, we previously showed that 24 h after tMCAO, the extent of brain damage varied in different brain regions. Thus, the lesion was primarily localized in the striatum of the right hemisphere of the rat brain [25]. Using RNA-Seq, we compared gene-expression profiles during ischemia in brain regions with different degrees of damage, namely, in the penumbra- and viable cell-associated FC and striatum that contained the infarct area and partly the penumbra area. As a result, more than one thousand genes both overlapped and were specific to FC and striatum at 24 h after tMCAO were identified [25]. Here, we analyzed striatal transcriptome of animals subjected to ischemia and receiving the peptides (Semax and ACTH(6-9)PGP). We hypothesized that in the striatum, as in the FC, peptides would be able to show similar effects tending to compensation of the gene-expression profile disrupted by ischemia. For hypothesis testing, we compared the expression profiles induced by the peptides in different tissues. We found that the number of genes overlapping between tissues was less than those that were in relative complements. In addition, there were genes that showed opposite directed changes in expression in the ischemic striatum and in FC under the influence of the same peptide. Moreover, there were almost 4 times fewer DEGs in the striatum than in FC. Thus, the transcriptomic effect of peptides in penumbra- and viable cell-associated FC was much more extensive than in the damaged striatum. This effect may be associated with the limited ability of striatal cells located in the area of severe damage to initiate regenerative mechanisms. Another feature was that the number of ischemia-induced DEGs in the striatum was, in contrast, greater than in the FC. Consequently, the peptides had a significantly lesser modulating effect on ischemia-induced genes in the striatum than in the FC.

An important characteristic of the effect of the Semax and ACTH(6-9)PGP peptides on the brain transcriptome during ischemia was that they could differentially modulate ischemia-induced gene-expression changes depending on the distance of the brain region from the ischemic focus. In FC containing penumbra cells, peptides significantly compensated for gene expression profiles after ischemia. Activation of inflammatory genes and reduction in the mRNA level of neurotransmission-related genes were observed under the influence of ischemia. Under the influence of peptides, on the contrary, a decrease in the expression of genes associated with inflammation and an activation of the expression of genes involved in neurotransmission were observed 24 h after tMCAO [23]. Using both RNA-Seq and real-time PCR on an expanded sample of animals, we showed that the peptides could reduce the expression of the *Hspb1*, *Cxcl16*, and *Casp3* genes in the FC containing the penumbra zone and had no significant effect on the expression of these genes in the striatum. Proteins encoded by genes are known to perform major functions in the cell. For example, the chemokine CXCL16 modulates neurotransmitter release in the hippocampal CA1 area [26]. The association of its expression with stroke has also been shown previously [27]. Additionally, CXCL16 coordinates MCP-1/CCL2 and adenosine A3 receptor activity to shield neurons in the CNS from excitotoxic cell death [28]. Long-term survival ischemia has been shown to induce neuronal death in CA3 via activation of caspase 3 [29]. Moreover, a decrease in the level of cleaved caspase-3 was shown under the action of morin in a model of heavy metal toxicity of PC12 cells [30]. The decrease in *Cxcl16* and *Casp3* gene expression under the influence of Semax and ACTH(6-9)PGP, observed in the present study, may be a potential node of the neuroprotective effect of the peptides in the penumbra. Also, ischemia was associated with an increase in the expression of the chaperone *Hspb1* gene both in FC and striatum. Such a result was also observed earlier, at both 4.5 and 24 h after tMCAO in rats [22,31]. However, there are studies showing that its involvement is more complex, and the chaperone may be involved in neuroprotection. Thus, reactive astrocytes secrete the chaperone HSPB1 to mediate neuroprotection [32]. Furthermore, *Hspb1* overexpression also ameliorates hypoxic-ischemic brain injury by reducing ferroptosis in rats via upregulating G6PD expression [33]. This may explain the spatial regulation of the expression of this gene under the influence of the peptides, which was observed 24 h after tMCAO in brain areas at different distances from the ischemic focus.

We found that the effect of Semax in the striatum was weaker than in the FC, but still in both tissues it was predominantly compensatory 24 h after tMCAO. In particular, the peptide affected 260 out of 269 overlapping DEGs related to the control of neurotransmission and the immune system in striatum. Concomitantly, ACTH(6-9)PGP peptide was also associated with compensation of the gene-expression profile impaired by IR. It was highlighted by 294 out of 446 ACTH(6-9)PGP-induced genes in striatum. But also peptide action was associated with enhancement of ischemia’s impact on gene-expression patterns in the striatum. Thus, we found that in the striatum, two related peptides (Semax and ACTH(6-9)PGP) generated both common and different effects at the transcriptome level. As is known, they had a common amino acid sequence in the ACTH(6-7) region and the C-terminal residue of PGP. At the same time, the ACTH(4-5) and ACTH(8-9) regions are specifically for Semax and ACTH(6-9)PGP, respectively. However, the effect of peptides is probably nonlinearly related to differences in chemical structure, since in different tissues with different degrees of damage, its effect manifests differently. It is possible that this property of the peptides can be used for target drugs on a specific area of the brain or on a focus of infarction of various localizations.

Functional analysis of DEGs under Semax and ACTH(6-9)PGP action revealed far fewer pathway associations in the striatum than in the FC. The pathways of PC1 were predominantly associated with both peptides’ impact on nerve impulse transmission. The other pathways were uniquely for Semax (PC2) and ACTH(6-9)PGP (PC3) effects in the striatum. The PC1 and PC2 pathways highlighted the effects of Semax as a compensatory one. The peptide predominantly decreased the expression of genes of the inflammatory cluster (IC) and increased the expression of genes of the neurotransmission cluster (NC) within the PC1-PC2. Ischemia had mostly the opposite effect on these systems.

Previously, we observed an effect that enhances the influence of ischemia on the genes of the inflammatory system for ACTH(6-9)PGP in the FC in the early hours after ischemia, 4.5 h after tMCAO. This suggested that there is a heterogenization of the inflammatory response into two branches, namely the enemy branch and the friend branch [22]. The enemy branch includes the proinflammatory response of the genome, which is mainly reduced by the peptide, and the friend branch consists of anti-inflammatory molecules, whose expression can be enhanced by the peptide for greater intensification of regenerative processes in the cell during ischemia. Here, a group of IC genes was seen, including *Actr3b*, *Mapk11*, and *Nfact2* (a fourth group of genes, Figure 5), whose expression was reduced by ACTH(6-9)PGP in the striatum 24 h after tMCAO. However, the peptide did not change the expression of genes of anti-inflammatory cytokines IL4 -10, -13, -17 and respective receptors in either the striatum or the cortex 24 h after tMCAO. The only exception was the IL10 receptor, the alpha (Il10ra) gene, the expression level of which was increased by the peptide in the striatum. Additionally, we noticed an increase in the expression of the *Junb* and *Irf7* genes, which could be important in neuroprotection [34,35]. Furthermore, and in accordance with literature data, an increase in the expression of genes such as *Tlr2*, *Tlr4*, *Olr1* (First gene group, Figure 5) points to the absence of a neuroprotective effect of the peptide in the striatum [36,37,38]. However, the result could have been a consequence of severe damage that the cell received from ischemia in this area of the brain.

The ACTH(6-9)PGP peptide also modulated the expression of NC genes in the striatum at 24 h after tMCAO. Interestingly, 294 genes whose expression changes were compensated by the peptide during ischemia were mainly clustered by functions of the neurosignaling system. These genes could indicate a neuroprotective effect of the peptide in the striatum. However, the effect of the ACTH(6-9)PGP on the neurotransmission genetic response was more complex. Namely, the peptide reduced the expression of many genes (e.g., *P2rx6*, *Gabra3*, *Grik4*, *Oprl1*, *Glra2*, *Crhr1*, *Hrh1*, *Chrm5*, *Grik1*) related to the neurosignaling system and whose expression was not significantly changed by ischemia itself. In particular, there were 289 genes that were DEGs in IA-s vs. IR-s but not in IR-s vs. SO-s at 24 h after tMCAO in striatum. Neuroactive ligand-receptor interaction was a key pathway annotation identified as significant for this gene set. Thus, the ACTH(6-9)PGP peptide had an effect on the genes of the neurosignaling system, which can be considered as a transcriptome trace of an ischemia-independent route of peptide action in the striatum.

It should be noted that among the DEGs under the influence of ACTH(6-9)PGP were many receptor genes, including *Gabra5*, *Gabra3*, *Htr2d*, *Grin2b*, *Grik4*, *Grik1,* and others (second gene group, Figure 5). The suppression of expression of these genes could be a consequence of severe irreversible damage instances in the striatum. Simultaneously, we observed an increase in the expression of a number of NC genes under the influence of ACTH(6-9)PGP (third gene group, Figure 5). Also, the genes were among those whose expression increased under the influence of ACTH(6-9)PGP in striatum at 24 h after tMCAO. Such an effect of ACTH(6-9)PGP on expression of *Adora2a*, *P2yr1* and *Gdnf* genes can be considered neuroprotective [39,40,41,42]. Thus, the ACTH(6-9)PGP peptide exhibits neuroprotective properties in the striatum, but its effect is probably greatly offset by severe damage to brain tissue.

The difference in the influence of peptides in different parts of the brain after ischemia may be associated with different mechanisms of cell death during ischemia. Thus, in the area of the ischemic nucleus, cells die mainly by the type of necrosis, but in the penumbra zone, the spectrum of cell death mechanisms is very diverse. It includes apoptosis, pyroptosis, necroptosis, PANoptosis, ferroptosis, autophagy, and even NETosis [43,44,45,46]. In FC, the proportion of penumbra cells is higher than in the striatum, in accordance with histological examination. Therefore, these mechanisms can be actively implemented in FC. In view of this, the transcriptomic response of peptides in FC is more extensive and diverse. In the striatum, due to extensive damage, the state of specific cells may play a greater role, which may contribute to the differences in the effects of the two peptides. In addition, the interaction of various cell death pathways plays an important role. In particular, inhibition of apoptosis can increase the risk of necroptosis. Also, mutual coordination of immune cells is observed during stroke. Thus, neutrophils are attracted to damaged brain tissue, which can aggravate inflammation. However, as a result, balanced regulation of these pathways can contribute to an increase in neuronal survival [43]. Peptides are substances regulating multiple organ functions [47]. They meet the needs of integrated correction of various cellular responses during stroke. At the same time, the features of their structure can impose restrictions on the stages of application or the degree of influence. In addition, peptides are able to provide allosteric interactions of the patterns of peptide metabolites (synactone) with various types of receptors [20]. As a result, signaling in the cell nucleus for transcriptome response through different signaling pathways (NC, IC, cell death, and others) can be modulated by peptides. In particular, this can explain the differences observed in the action of ACTH-like peptides with neuroprotective potential in rat brain regions with different degrees of ischemic damage.

As we previously revealed, the molecular mechanisms of the Semax and ACTH(6-9)PGP peptide action include the compensation of the rat brain gene-expression profile of FC cells a day following tMCAO [23]. Concomitantly, here we revealed traces of the neuroprotective effect of Semax and ACTH(6-9)PGP in the striatum at 24 h after tMCAO. They may be valuable for achieving the spatial and temporal specificity required for therapeutic interventions after ischemia. Recently, single-cell RNA sequencing (scRNA-Seq) identified a critical role for leukocyte immunoglobulin-like receptor B4 (LILRB4) in microglia in modulating the immune response after stroke by regulating CD8+ T cell infiltration and activation. Moreover, LILRB4 knockout aggravates ischemic brain damage, while LILRB4 overexpression, on the contrary, provides neuroprotection [48]. It should be noted that inhibition/overexpression of key hub genes that we identified in the networks (Figure 5) appears to be the most promising to study the effects of ACTH-like peptides in ischemic brain cells at different distances from the lesion while taking into account the anatomy of the brain. Among the candidates, we consider *Fos*, *Tlr4*, *Pik3cd*, *Mapk11*, and *Nfatc2* genes, which had the most connections each with IC, as well as *Cacna1a*, *Grin2d*, *Gabbr2*, *Gabra3*, *Gabra5 Adora2a*, and *Prkcg* genes, which had the most connections each with NC. As a result, methods of spatial transcriptomics may reveal the enigma of spatial regulation of gene expression in the brain and explain the reasons for the compensatory effect of neuroprotectors in some areas of the brain and a more disturbing effect in others. Further protein studies would be significant for translating genome effects into phenotype meaning and into clinical application as well. We believe that identifying genome and metabolic responses, not only locally but also considering the entire brain tissue, can be key for successfully inducting the regeneration processes in brain cells after stroke.

## 4. Materials and Methods

### 4.1. Animals

The Wistar line of white male rats, weighing between 200 and 250 g, were acquired from AlCondi, Ltd. (Moscow, Russia), as previously mentioned [25]. Four groups of animals were formed: “sham operation” (SO), “ischemia-reperfusion” (IR), “ischemia-reperfusion + Semax” (IS), and “ischemia-reperfusion + ACTH(6-9)PGP” (IA). The transient middle cerebral artery occlusion (tMCAO) model was applied to IR, IS, and IA animals. At least five animals were included in each experimental group.

### 4.2. tMCAO Model

The tMCAO model with 90 min occlusion was carried out using Koizumi et al.’s methodology [49] under histological examination (HE) and magnetic resonance imaging (MRI) as previously detailed [23,24]. Appendix A provide descriptions of the tMCAO and HE details, respectively. Animals underwent operation, occlusion, MRI, and decapitation under temporal isoflurane anesthesia. At 24 h after the beginning of the tMCAO/sham operation, the rats were decapitated.

In addition, the rats received intraperitoneal administration of peptide (Semax or ACTH(6-9)PGP) at 1.5 h, 2.5 h, and 6.5 h after the beginning of the surgical operation (tMCAO) at doses of 100 µg/kg rat weight, respectively, in compliance with [18,50,51]. Thus, the groups IS and IA after Semax and ACTH(6-9)PGP were, respectively, formed. The animals of the IR and SO groups received saline injections simultaneously.

### 4.3. Sample Collection and RNA Isolation

The same animals were used to collect the ipsilateral striatum and ipsilateral fragments of the frontal cortex (FC) area at +2 to +5 mm from the bregma. SO-f, IR-f, IS-f, and IA-f were the names of the FC samples of the SO, IR, IS, and IA groups, whereas SO-s, IR-s, IS-s, and IA-s were the names of the striatum samples, respectively. Every sample was stored at −70 °C after being in an RNAlater (Ambion, Austin, TX, USA) solution for 24 h at 4 °C. Following the isolation of total RNA, capillary electrophoresis (Experion, BioRad, Hercules, CA, USA) was used to verify the integrity of the RNA, as previously mentioned [18]. At least 9.0 was the RNA integrity number (RIN).

### 4.4. RNA-Seq

As previously described [18], RNA-Seq analysis was performed using an Illumina HiSeq 1500 device (Illumina, San Diego, CA, USA) to determine the polyA fraction of the total RNA. The number of generated reads was at least 10 million (1/50 nt). The RNA-Seq analysis was performed with the participation of OOO Genoanalytika, Moscow, Russia.

### 4.5. cDNA Synthesis and Real-Time Reverse Transcription Polymerase Chain Reaction (RT-PCR)

cDNA was created, as described in the past [18], using oligo (dT)18 primers. The Evrogen Joint Stock Company manufactured the PCR primers, which were selected using the OLIGO Primer Analysis Software version 6.31 (Appendix A). As previously mentioned [18], each cDNA sample was examined three times using RT-PCR.

### 4.6. RNA-Seq Data Analysis

Each of the comparison groups (SO, IR, IS, and IA) for the RNA-Seq experiments contained three animals (n = 3). As previously mentioned [18], Cuffdiff/Cufflinks software v.2.2.1 was utilized for gene annotations. The Cuffdiff tool was used to quantify the levels of mRNA expression, as previously described [18]. Only genes with *p*-values (*t*-test) adjusted using the Benjamini-Hochberg approach below 0.05 (*Padj* < 0.05) and >1.5-fold changes in expression were taken into consideration.

### 4.7. Real-Time RT-PCR Data Analysis

The relative gene expression was calculated using Relative Expression Software Tool (REST) (v. 2005) software (gene-quantification, Freising-Weihenstephan, Bavaria, Germany) [52,53], as previously described [18]. The values were computed as Ef^Ct(ref)^/Ef^Ct(tar)^, where Ef is the PCR efficiency, Ct(tar) is the target gene’s average threshold cycle (Ct), Ct(ref) is the reference gene’s average Ct, and Ef^Ct(ref)^ is the reference genes’ geometric average Ef^Ct^. All PCR reactions had efficiency values between 1.91 and 2.11 (Appendix A). The expression of the cDNA samples was normalized using the reference gene *Gapdh*. The comparison groups had five animals each. Significant differences were considered at *p* < 0.05.

### 4.8. Functional Analysis

The Database for Annotation, Visualization and Integrated Discovery (DAVID (2021 Update) [54]) was used to annotate the functions of the differentially expressed mRNAs (DEGs), as well as Gene Set Enrichment Analysis (GSEA) [55]. Only functional categories with *Padj* < 0.05 were taken into consideration. The DEGs were subjected to hierarchical cluster analysis using Heatmapper (Wishart Research Group, University of Alberta, Ottawa, Canada) [56]. Using Microsoft Excel (Microsoft Office 2010), a volcano plot was created. The regulatory network was visualized using Cytoscape 3.9.2 software (Institute for Systems Biology, Seattle, WA, USA) [57].

### 4.9. Availability of Data and Material

The RNA-sequencing data were deposited in the Sequence Read Archive database under accession code PRJNA1119923 (SAMN41664920-SAMN41664931, https://dataview.ncbi.nlm.nih.gov/object/PRJNA1119923?reviewer=plp0lftpig94lmq8vmup84at3d (accessed on 25 June 2024)) [58], PRJNA1128447 (SAMN42050762-SAMN42050773, https://dataview.ncbi.nlm.nih.gov/object/PRJNA1128447?reviewer=vkci32ofqame6mkpbivjt1pqn8 (accessed on 25 June 2024)) [59].

## 5. Conclusions

In conclusion, genes that are associated with the action of ACTH-like peptides (Semax and ACTH(6-9)PGP) in areas of the rat brain with varying degrees of ischemia injury were identified. At 24 h after tMCAO, the transcriptomic effect of peptides in the penumbra-associated FC was much more extensive than in the infarct-associated striatum. Moreover, differential spatial regulation of the ischemia process in the rat brain at the transcriptome level was discovered under peptides with different ACTH structures. In FC, both peptides exhibited a pronounced compensatory effect upon the disturbances in the gene-expression profile after ischemia. In striatum, the effect of Semax was predominantly compensatory and aimed at reducing the effect of ischemia. Concomitantly, ACTH(6-9)PGP peptide was able to enhance the effect of ischemia on many genes, in addition to the compensatory effect. These genes were primarily associated with inflammation. Hence, ACTH(6-9)PGP peptide exhibited a heterogeneous effect in the striatum. And so our study identified genes which are subject to a highly variable expression within different parts of the damaged hemisphere under the influence of the neuroprotective peptides. We believe that the transcriptome responses of the identified genes can be the key for efficiently inducting regeneration processes in brain cells after stroke. Furthermore, our results may be useful for selecting more effective neuroprotective drug structures in accordance with their specific tissue/damage therapeutic impact.

## Figures and Tables

**Figure 1 ijms-26-06256-f001:**
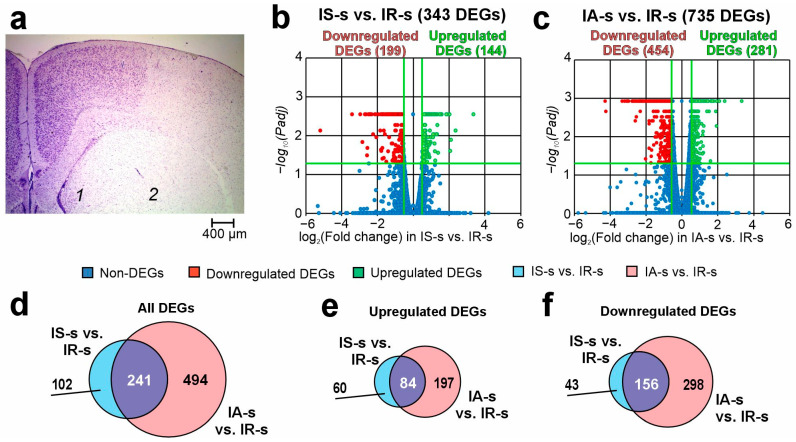
RNA-Seq analysis of the effect of Semax and ACTH(6-9)PGP on the transcriptome of damaged striatum of rats 24 h after tMCAO. (**a**) Serial coronal brain section at the levels of +0.48 mm from the bregma. The right (ipsilateral) hemisphere is shown. Zone of penumbra and normal tissues of the striatum is shown by “1”, whereas zone of necrotic tissue is shown by “2”. (**b**,**c**) Volcano plots of RNA-Seq results show a comparison of the gene distribution between the IS-s and IR-s groups (**b**), as well as IA-s and IR-s (**c**). Upregulated and downregulated DEGs are represented as red and green dots, respectively (fold change > 1.50; *Padj* < 0.05). Not differentially expressed genes (non-DEGs) are represented as blue dots (fold change ≤ 1.50; *Padj* ≥ 0.05). (**d**–**f**) Venn diagrams represent results obtained in comparisons between the IS-s vs. IR-s and IA-s vs. IR-s. All (**d**), upregulated (**e**), and downregulated (**f**) DEGs are shown for comparison.

**Figure 2 ijms-26-06256-f002:**
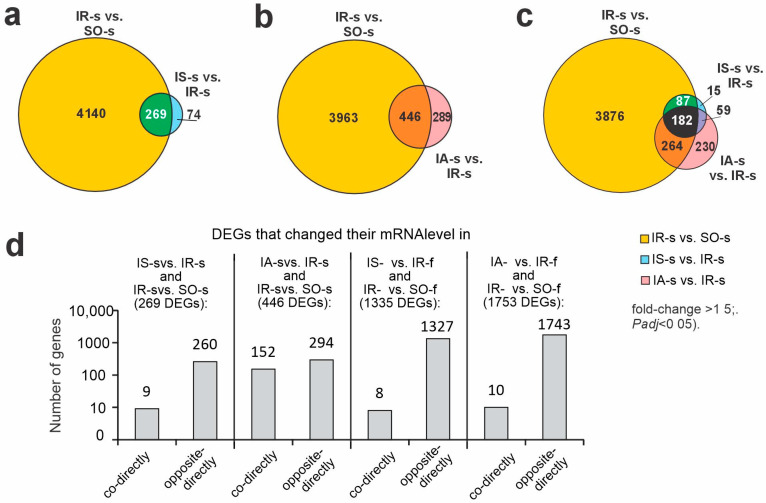
Differences in rat brain transcriptomes following ischemia and after Semax and ACTH(6-9)PGP administration in striatum and FC at 24 h after tMCAO. (**a**–**c**) Venn diagrams represent results obtained in comparisons between the IR-s vs. SO-s and IS-s vs. IR-s (**a**), IR-s vs. SO-s and IA-s vs. IR-s (**b**), as well as IR-s vs. SO-s, IS-s vs. IR-s and IA-s vs. IR-s (**c**) groups in striatum. (**d**) Diagram showing common DEGs between peptides and IR action in striatum and FC.

**Figure 3 ijms-26-06256-f003:**
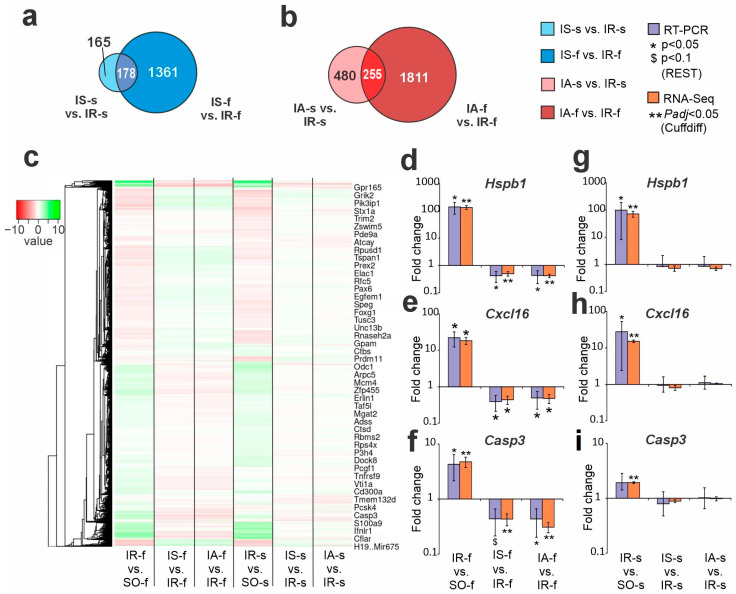
Comparative gene-expression results for IR, Semax and ACTH(6-9)PGP action in striatum and FC at 24 h after tMCAO. (**a**) Venn diagrams represent results obtained from comparing the IS-s vs. IR-s in striatum and IS-f vs. IR-f in FC. (**b**) Venn diagrams represent results obtained from comparing the IA-s vs. IR-s in striatum and IA-f vs. IR-f in FC. (**c**) Hierarchical cluster analysis of all DEGs in IR-f vs. SO-f, IS-f vs. IR-f, IA-f vs. IR-f, IR-s vs. SO-s, IS-s vs. IR-s, and IA-s vs. IR-s, where each row represents a DEG. As a significant result, only genes with cut-off >1.5 and *Padj* < 0.05 were chosen. (**d**–**i**) Real-time RT-PCR studies the expression patterns of *Hspb1*, *Cxcl16,* and *Casp3* genes under IR, Semax, and ACTH(6-9)PGP action in FC (**d**–**f**) and striatum (**g**–**i**), respectively. The expression of the cDNA samples was normalized using the reference gene *Gapdh*. PCR calculations were performed using the Relative Expression Software Tool (REST) 2005 software. When comparing groups of PCR data, differences with a *p*-value less than 0.05 (two-sided Pair-Wise Fixed Reallocation Randomization Test) were considered as a significant. RNA-Seq results are presented for comparison. *Padj* is the *p*-value (*t*-test) adjusted with the Benjamini-Hochberg correction.

**Figure 4 ijms-26-06256-f004:**
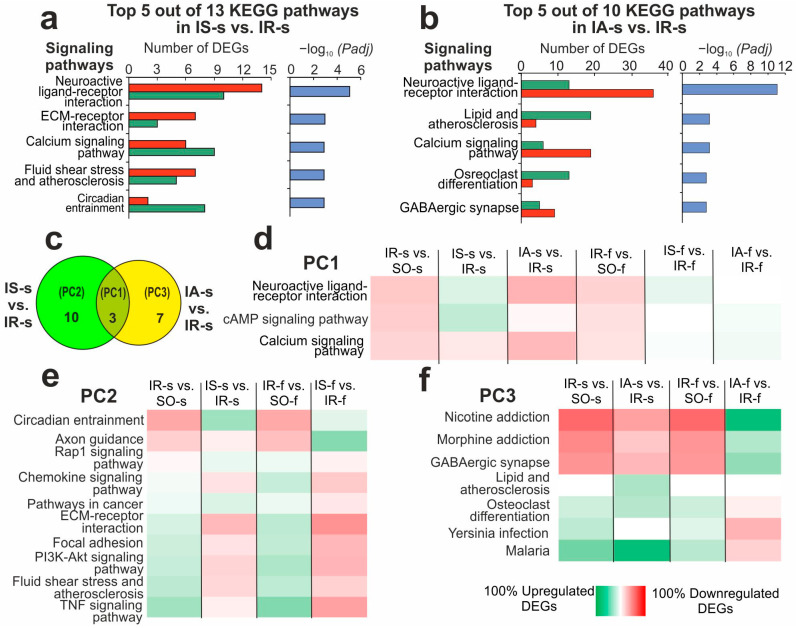
Functional annotations of DEGs associated with Semax and ACTH(6-9)PGP peptide action in striatum of rats at 24 h after tMCAO. (**a**,**b**) The top 5 most significant KEGG pathways among the annotations in each comparison group and the corresponding *Padj* values, as well as the number of upregulated (green) and downregulated (red) DEGs in the two pairwise comparisons IS-s vs. IR-s (**a**) and IA-s vs. IR-s (**b**) are presented. The pathway-enrichment analysis of DEGs was carried out according to DAVID (2021 update). (**c**) Schematic comparison of DEG-related annotations in IS-s vs. IR-s and IA-s vs. IR-s pairwise comparisons in striatum presented using a Venn diagram. Numbers on the chart segments represent the number of annotations. (**d**–**f**) Heatmap of overlapped (**d**), as well as unique signaling pathways associated with DEGs in IS-s vs. IR-s (**e**) and IA-s vs. IR-s (**f**) pairwise comparisons in striatum. A pairwise comparison is represented by each column, and signaling pathway is represented by each row (KEGG). The pathways with which the majority of upregulated genes are associated are represented by green bars, and the pathways with which the majority of downregulated genes are associated are represented by red bars. Only DEGs and pathways with *Padj* < 0.05 were selected for analysis; *n* = 3 animals per group.

**Figure 5 ijms-26-06256-f005:**
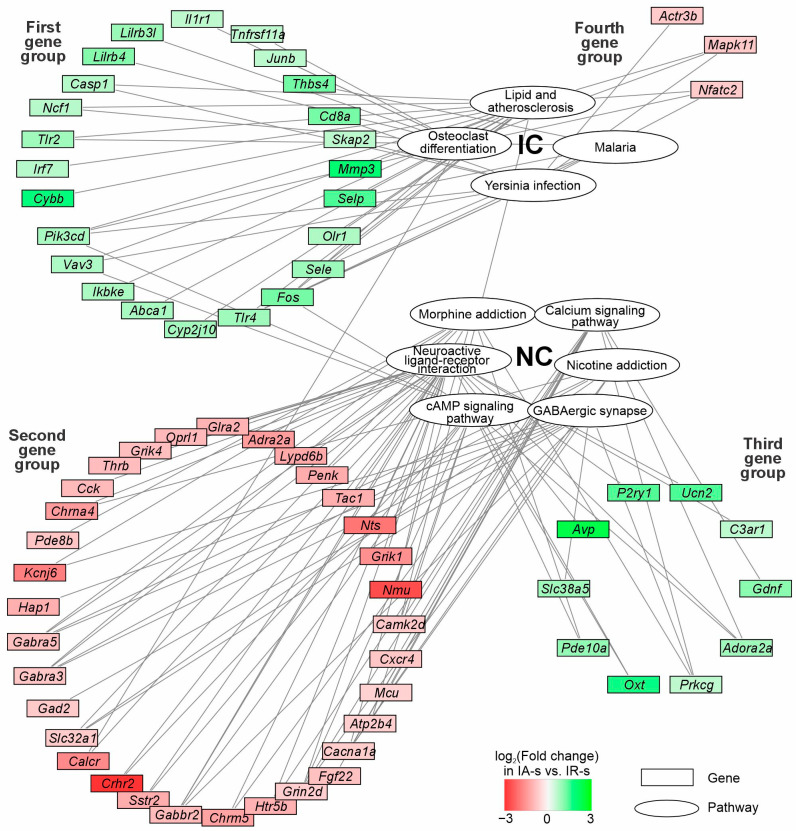
The involvement of the DEGs of NC and IC associated with effects of ACTH(6-9)PGP peptide a day after IR conditions in striatum. In the network, pathway clusters (NC and IC) are grouped in the white ellipses. The genes are represented by rectangular blocks colored according to their differential expression in IA-s vs. IR-s pairwise comparison. The only genes included were in the network were DEGs in IA-s vs. IR-s but not IA-f vs. IR-f at 24 h after tMCAO. The lines connecting the pathways and genes indicate the participation of the protein products of genes in the pathway functioning. Genes are grouped in accordance with their association with pathway clusters, as well as up/downregulation of their expression in IA-s vs. IR-s. Thus, four gene groups were revealed in the network. The KEGG databases from the DAVID database (2021 update) were used to annotate all clustered signaling pathways. Only DEGs and annotations with *Padj* < 0.05 were considered to be significant. Cytoscape 3.9.2 (Institute for Systems Biology, Seattle, WA, USA) was used to create the network.

## Data Availability

Publicly available datasets were analyzed in this study. These data can be found here: [60].

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
