# Peer review of "Genes That Associated with Action of ACTH-like Peptides with Neuroprotective Potential in Rat Brain Regions with Different Degrees of Ischemic Damage"

_ijms, 2025, doi:10.3390/ijms26136256_

Round 1
Reviewer 1 Report
Comments and Suggestions for Authors
Although the objective of the work is well read in the abstract, it lacks power in the way it is mentioned in the Introduction. I suggest some changes to draw the attention of the reader to the main objective of the work in the line 81. Overall, the work is very well constructed and explored. Methodology is well described and/or referenced. Results are well worked and presented, however there are corrections to be done in some descriptions to make the work understandable. The same for the discussion and conclusions. Though the discussion’s rationale is well built, and the conclusions are in line with the results, I recommend some corrections and/or paraphrasing to successfully convey the very much pertinent message found in this scientific work. Here I share a point-by-point review of this manuscript:
Please verify Supplementary Figure S2. Did you mean “8 DEGs that opposite-directly changed their m-RNA level in IS-s vs IR-s and IS-s vs IR-s” OR did you mean “8 DEGs that opposite-directly changed their m-RNA level in IS-s vs IR-s and IS-f vs IR-f? If you meant IS-f vs IR-f, please correct it in the slide.
Please verify Supplementary Figure S4. For a better understanding, I suggest correcting the following text: “(a, b) Venn diagrams shows a comparison of sets pathways between four IR-s vs. SO-s, IS-s vs. IR-s, IR-f vs. SO-f and IS-f vs. IR-f pairwise comparisons” to “Venn diagrams show a comparison of pathways sets between four IR-s vs. SO-s, IS-s vs. IR-s, IR-f vs. SO-f and IS-f vs. IR-f pairwise comparisons”.
Please verify Supplementary Method S2. I suggest aligning all the test to justified.
Please verify the main manuscript. Line 34: "Both peptides tended to normalised profile", please correct to " Both peptides tended to a normalized profile..."
Line 35:" whereas 152 genes showed even more disturbing profile ". Please correct to: "whereas 152 genes showed an even more affected profile"
Lines 43-44:"in accordance with their tissues- and damage-specific therapeutic impact". I suggest using: " in accordance with their specific-tissue/damage therapeutic impact".
Line 49: " that leads to". I suggest: "that usually leads to"
Line 61: " studied spectrum of neuroprotective activity". Please correct to: "studied the spectrum of neuroprotective activity"
Line 62: " effect to ischemic brain cells ". I kindly suggest: "effect in ischemic brain cells".
Line 72: " a neurotransmitter clusters". Please correct to " a neurotransmitter cluster".
Line 81: to better underline the objective in this study I kindly suggest altering “using RNA-Seq we studied the effect of Semax and ACTH(6–9)PGP peptide” to “ using RNA-Seq we aimed to study the effect of Semax and ACTH(6–9)PGP peptide”
Line 92: " that showed even more disturbing profile". I kindly suggest: " that showed an even more affected profile"
Line 93-94: "These DEGs were associated with inflammation, predominantly." I kindly suggest: "These DEGs were predominantly associated with inflammation."
Line 94: " About hundred genes were ". I suggest correcting to: " About one hundred genes were"
Line 101: " their tissues- and damage-specific therapeutic impact." I kindly suggest: "with their specific-tissue/damage therapeutic impact".
Line 171: Please correct: “file caused by ischemia predominantly.” To “file predominantly caused by ischemia.”
Line 188: I kindly suggest correcting “showed even more disturbing profile after ACTH(6-9)PG” to “showed an even more affected profile after ACTH(6-9)PG”
Line 196: Please add the number of the figure where these results can be found at the end of the sentence:”(…) but not IA-s vs. IR-s pairwise comparison (Figure 2c).”
Line 200: Please add the number of the figure where these results can be found at the end of the sentence: “IR-s pairwise comparisons (Figure 2c).”
Line 216: I kindly suggest changing “Venn diagrams are evident that” to “ Venn diagrams clearly evidence that”
Lines 220-221: I kindly suggest changing “Overlapping genes co-directly changed their mRNA level in both tissues at 24 h tMCAO, predominantly” to “ Overlapping genes predominantly co-directly changed their mRNA level in both tissues at 24 h tMCAO.”
Line 238: I kindly suggest changing “and were associated with Neuroactive ligand-receptor interaction, predominantly.” To “and were predominantly associated with Neuroactive ligand-receptor interaction.”
Line 242-243: I kindly suggest changing “in the two tissues studied” to “ for both studied tissues.”
Line 244: Please change “and increased” to “but increased”.
Line 245: Please change “was observed more rarely” to “ was more rarely observed.”
Line 248: Please remove “ in real time” after the acronym (RT–PCR) in: “Real-time reverse transcription polymerase chain reaction (RT–PCR) in real time”.
Line 250: Please add spaces in between commas and letters in: ”(Figure 3d,e,f) and FC (Figure 3g,h,k)”.
Line 254: Please add spaces in between commas and letters in: “(Figure 3d,e,f)”.
Line 255: Please add spaces in between commas and letters in: “(Figure 3g,h,k)”.
Line 258: Please simplify “obtained in comparisons between” to “obtained from comparing”
Line 259-260: Please simplify “obtained in comparisons between the” to “ obtained from comparing the”.
Line 284: Please change “of sets pathways” to “ of pathways sets”.
Line 287: Please correct “As a result, more than hundred pathways were” to “As a result, more than a hundred pathways were” or “As a result, more than one hundred pathways were”.
Line 295: Please correct “This pathway group we named by Pathway Cluster 1” to “ This pathway group was denominated Pathway Cluster 1”.
Line 297: I kindly suggest changing “regardless FC or striatum” to “regardless being FC or striatum tissue”.
Line 300: I kindly suggest changing “ there were 10 and 7 pathways” to “ there were identified 10 and 7 pathways”.
Line 301: I kindly suggest simplifying “They were named by” to “They were named”.
Lines 307: Please correct “Semax predominantly compensate of” to “Semax predominantly compensates for”.
Line 312: Please correct “genes were associated with NC” to “289 genes associated to NC”.
Line 314: Please correct “We calculated number of up- and (…)” to “ We calculated a number of 289 of up- and (…).”
Line 316: Please remove 289 gene from :“289 gene, Figure 2e, Supplementary Table S11)”.
Line 317: Please correct “as from overlapping genes“ to “as 294 genes overlapping genes”.
Lines 317-318: Please remover 294 genes from: “(294 genes, Figure 2b)”.
Line 318: Please correct “and co-directly” to “and 152 genes co-directly”.
Line 318: Please remove 152 genes from: “(152 genes, Figure 2b)”.
Line 336: Please correct “ Top 5 the most significant” to “Top 5 of the most significant”.
Line 363: Please correct “In the scheme, pathway clusters (NC and IC) are grouped in the white” to “The scheme shows pathway clusters (NC and IC) grouped in white”.
Line 369: Please correct “with pathway clusters” to “ to pathway clusters”.
Line 390: Please correct “ associated with effects of” to “associated to effects of”.
Line 393: Please correct “Only those genes were included” to “The only genes included were”
Line 394: Please remove the word that from: “network that were DEGs”
Lines 407-408: Please correct “those of them that reduced expression” to “those which reduced expression”.
Line 408: Please correct “vice versa” to “vice-versa”.
Line 409: Please correct “those of them that increased expression” to “ those which increased expression”.
Line 416: Please correct “than thousand genes” to “than one thousand genes”.
Line 416: Please correct “and specific to FC” to “ and were specific to FC”.
Line 422: Please remove “that were” from: “the number of genes that were overlapping between”.
Line 435: Please correct “In the frontal cortex” to “ In FC”.
Line 437: Please correct “ (…)ischemia, activation of genes involved(…)” to “ischemia, it was observed activation of genes involved (…)”
Line 438: Please remove “were observed” from “with neurotransmission were observed”.
Line 441: Please include a comma in: “Using both RNA(…)” as in “Using both, RNA (…)”.
Line 441: Please include a comma in “real-time PCR on an(…)”, as in “real-time PCR, on an (…)”
Line 453: Please remove “by us” in “observed by us in the present study (…)”.
Line 456: Please correct: “Such a result we also observed earlier both” to “Such a result we was also observed earlier, at both(…)”.
Line 462: Please correct “ which we observed 24 hours after tMCAO (…)” to “which we was observed 24 hours after tMCAO(…)”.
Line 472: Please insert a comma in “generated both common (…)” as follows “generated both, common (…)”.
Line 477: Please correct “the effect of peptides manifests itself differently (…)” to “the effect of peptides its effect manifests itself differently (…)”.
Line 478: Please correct “can be used in studying the targeted effect of drugs(…) “ to “can be used in studying the targeted effect of for target drugs (…)”.
Line 482: Please correct “(…) associated with both peptides impact (..)” to “predominantly associated with both, peptides impact(..)”.
Line 482: Please remove predominantly from the end of the sentence: “(…)nerve impulse transmission, predominantly.”
Line 487: Please correct “Ischemia had predominantly the opposite effect(…)” to “Ischemia had mostly the opposite effect (…)”.
Line 490: Please correct “We then suggested that there is(…)” to “It suggested that there is (…)”.
Line 493: I kindly suggest correcting “which is predominantly reduced by the peptide(…)” to “which is mainly reduced by the peptide (…)”.
Line 494: I kindly suggest correcting “the expression of which can be enhanced (…) to “ whose expression can be enhanced(…)”.
Lines 495-496: Please correct “Here, we observed a group of IC genes, (…)” to “Here, it was seen a group of IC genes, (…)”.
Line 499: Please correct “and their receptors(…)” to “and respective receptors (…)”.
Line 500: I kindly suggest using “was IL-10 receptor” instead of “was interleukin 10 receptor(…)”.
Lines 503-505: I kindly suggest using “Furthermore, and in accordance with literature data, an increase in the expression of genes such as Tlr2, Tlr4, Olr1 (First gene group, Figure 5) points to the absence of a neuroprotective effect of the peptide in the striatum [39–41].” instead of “Furthermore, an increase in the expression of genes such as Tlr2, Tlr4, Olr1 (First gene group, Figure 5) indicated the absence of a neuroprotective effect of the peptide in the striatum, in accordance with literature data. [39–41].”
Line 508: Please correct “also modulated expression of NC genes(…)” to “also modulated the expression of NC genes (…)”.
Line 550: Please correct “can explain the differences we observed in action of(…)” to “can explain the differences observed in action of (…)”.
Line 565: I kindly suggest correcting “from the lesion and taking into account(…)” to “ from the lesion while taking into account (…)”.
Lines 571-575: I kindly suggest paraphrasing the paragraph “Furthermore, further protein studies very significant to translate genome effects to phenotyping and clinical level. We believe that genome and metabolic responses identified not only local but entire brain can be the key for induction of regeneration processes in brain cells after stroke.” My humble suggestion for a more understandable reading:
“ Further protein studies would be significant for translating genome effects into phenotype meaning and into clinical application as well. We believe that identifying genome and metabolic responses, not only locally but also considering the entire brain tissue, can be key for successfully inducting the regeneration processes in brain cells after stroke.”
Line 586: "Koizumi et al. ". Please correct to "Koizumi et al.,”
Line 594: “respectively, were formed.”. Please correct to: “ were respectively formed.”
Line 601: “Then total RNA (…)”. Please correct to: “Then, total RNA (…)”
Line 620: “as previously described”. To avoid repetition, kindly suggest: “as described in the past”.
Line 650: Please correct “genes that associated with action of ACTH-like peptides(…)” to “genes associated to the action of ACTH-like peptides (…)”.
Line 656: Please correct “compensatory effect on the (…)” to “compensatory effect upon the (…)”.
Lines 660-661: Please correct “peptide exhibit a heterogeneous effect (…)” to “peptide exhibited a heterogeneous effect (…)”.
Lines 661-663: I suggest paraphrasing the sentence “Thus, we revealed genes, which are subject to the greatest variability in changes in expression in different parts of the damaged hemisphere under the influence of the neuroprotective peptides.” I humbly leave a suggestion:
“And so, our study identified genes which are subject to a highly variable expression within different parts of the damaged hemisphere under the influence of the neuroprotective peptides.”
Line 663-665: I kindly suggest paraphrasing the sentence “We believe that transcriptome responses of such genes identified can be the key for induction of regeneration processes in brain cells after stroke.” to “ We believe that the transcriptome responses of the identified genes can be the key for efficiently inducting regeneration processes in brain cells after stroke.”
Line 666: Please correct “tissues- and damage-specific” to “ specific-tissue/damage”.
Congratulations for your work and wishes of the best outcomes in your future endeavors.
Comments on the Quality of English LanguageI recommend an overall revision of the English language.
Author Response
Response to the comments of Reviewer 1 to Manuscript ID: ijms-3685289
Authors:
We are very grateful to the Reviewer 1 for the review and constructive comments. We carefully considered the comments of the Reviewer 1 and attached the answers to all comments.
Reviewer 1:
- Please verify Supplementary Figure S2. Did you mean “8 DEGs that opposite-directly changed their m-RNA level in IS-s vs IR-s and IS-s vs IR-s” OR did you mean “8 DEGs that opposite-directly changed their m-RNA level in IS-s vs IR-s and IS-f vs IR-f? If you meant IS-f vs IR-f, please correct it in the slide.
Authors:
In accordance with the Reviewer’s recommendation, changes were added in the Supplementary Figure S2.
Reviewer 1:
- Please verify Supplementary Figure S4. For a better understanding, I suggest correcting the following text: “(a, b) Venn diagrams shows a comparison of sets pathways between four IR-s vs. SO-s, IS-s vs. IR-s, IR-f vs. SO-f and IS-f vs. IR-f pairwise comparisons” to “Venn diagrams show a comparison of pathways sets between four IR-s vs. SO-s, IS-s vs. IR-s, IR-f vs. SO-f and IS-f vs. IR-f pairwise comparisons”.
Authors:
In accordance with the Reviewer’s recommendation, changes were added in the Supplementary Figure S4.
Reviewer 1:
- Please verify Supplementary Method S2. I suggest aligning all the test to justified.
Authors:
In accordance with the Reviewer’s recommendation, changes were added in the Supplementary Method S2. The magnification level ×2.5 was indicated.
Reviewer 1:
- Please verify the main manuscript. Line 34: "Both peptides tended to normalised profile", please correct to " Both peptides tended to a normalized profile...".
Authors:
We are grateful to the Reviewer for the comment. The changes were added in the text of the Manuscript (line 35 in Mark-up Copy_R1).
Reviewer 1:
- Line 35:" whereas 152 genes showed even more disturbing profile ". Please correct to: "whereas 152 genes showed an even more affected profile".
Authors:
In accordance with the Reviewer’s recommendation, changes were added in the text of the Manuscript (line 36 in Mark-up Copy_R1).
Reviewer 1:
- Lines 43-44:"in accordance with their tissues- and damage-specific therapeutic impact". I suggest using: " in accordance with their specific-tissue/damage therapeutic impact".
Authors:
In accordance with the Reviewer’s recommendation, changes were added in the text of the Manuscript (line 45 in Mark-up Copy_R1).
Reviewer 1:
- Line 49: " that leads to". I suggest: "that usually leads to".
Authors:
We are grateful to the Reviewer for the comment. The changes were added in the text of the Manuscript (line 49 in Mark-up Copy_R1).
Reviewer 1:
- Line 61: " studied spectrum of neuroprotective activity". Please correct to: "studied the spectrum of neuroprotective activity".
Authors:
In accordance with the Reviewer’s recommendation, changes were added in the text of the Manuscript (line 65 in Mark-up Copy_R1).
Reviewer 1:
- Line 62: " effect to ischemic brain cells ". I kindly suggest: "effect in ischemic brain cells".
Authors:
In accordance with the Reviewer’s recommendation, changes were added in the text of the Manuscript (line 67 in Mark-up Copy_R1).
Reviewer 1:
- Line 72: " a neurotransmitter clusters". Please correct to " a neurotransmitter cluster".
Authors:
In accordance with the Reviewer’s recommendation, changes were added in the text of the Manuscript (line 98 in Mark-up Copy_R1).
Reviewer 1:
- Line 81: to better underline the objective in this study I kindly suggest altering “using RNA-Seq we studied the effect of Semax and ACTH(6–9)PGP peptide” to “ using RNA-Seq we aimed to study the effect of Semax and ACTH(6–9)PGP peptide”.
Authors:
In accordance with the Reviewer’s recommendation, changes were added in the text of the Manuscript (line 105 in Mark-up Copy_R1).
Reviewer 1:
- Line 92: " that showed even more disturbing profile". I kindly suggest: " that showed an even more affected profile".
Authors:
In accordance with the Reviewer’s recommendation, changes were added in the text of the Manuscript (lines 121-122 in Mark-up Copy_R1).
Reviewer 1:
- Line 93-94: "These DEGs were associated with inflammation, predominantly." I kindly suggest: "These DEGs were predominantly associated with inflammation.".
Authors:
In accordance with the Reviewer’s recommendation, changes were added in the text of the Manuscript (line 123 in Mark-up Copy_R1).
Reviewer 1:
- Line 94: " About hundred genes were ". I suggest correcting to: " About one hundred genes were".
Authors:
In accordance with the Reviewer’s recommendation, changes were added in the text of the Manuscript (line 123 in Mark-up Copy_R1).
Reviewer 1:
- Line 101: " their tissues- and damage-specific therapeutic impact." I kindly suggest: "with their specific-tissue/damage therapeutic impact".
Authors:
In accordance with the Reviewer’s recommendation, changes were added in the text of the Manuscript (line 132 in Mark-up Copy_R1).
Reviewer 1:
- Line 171: Please correct: “file caused by ischemia predominantly.” To “file predominantly caused by ischemia.”.
Authors:
In accordance with the Reviewer’s recommendation, changes were added in the text of the Manuscript (lines 215-216 in Mark-up Copy_R1).
Reviewer 1:
- Line 188: I kindly suggest correcting “showed even more disturbing profile after ACTH(6-9)PG” to “showed an even more affected profile after ACTH(6-9)PG”.
Authors:
In accordance with the Reviewer’s recommendation, changes were added in the text of the Manuscript (line 233 in Mark-up Copy_R1).
Reviewer 1:
- Line 196: Please add the number of the figure where these results can be found at the end of the sentence:” (…) but not IA-s vs. IR-s pairwise comparison (Figure 2c).”.
Authors:
In accordance with the Reviewer’s recommendation, changes were added in the text of the Manuscript (line 241 in Mark-up Copy_R1).
Reviewer 1:
- Line 200: Please add the number of the figure where these results can be found at the end of the sentence: “IR-s pairwise comparisons (Figure 2c).”.
Authors:
In accordance with the Reviewer’s recommendation, changes were added in the text of the Manuscript (line 245 in Mark-up Copy_R1).
Reviewer 1:
- Line 216: I kindly suggest changing “Venn diagrams are evident that” to “Venn diagrams clearly evidence that”.
Authors:
In accordance with the Reviewer’s recommendation, changes were added in the text of the Manuscript (line 263 in Mark-up Copy_R1).
Reviewer 1:
- Lines 220-221: I kindly suggest changing “Overlapping genes co-directly changed their mRNA level in both tissues at 24 h tMCAO, predominantly” to “Overlapping genes predominantly co-directly changed their mRNA level in both tissues at 24 h tMCAO.”.
Authors:
In accordance with the Reviewer’s recommendation, changes were added in the text of the Manuscript (lines 220-221 in Mark-up Copy_R1).
Reviewer 1:
- Line 238: I kindly suggest changing “and were associated with Neuroactive ligand-receptor interaction, predominantly.” To “and were predominantly associated with Neuroactive ligand-receptor interaction.”.
Authors:
In accordance with the Reviewer’s recommendation, changes were added in the text of the Manuscript (lines 268-270 in Mark-up Copy_R1).
Reviewer 1:
- Line 242-243: I kindly suggest changing “in the two tissues studied” to “for both studied tissues.”.
Authors:
In accordance with the Reviewer’s recommendation, changes were added in the text of the Manuscript (line 292 in Mark-up Copy_R1).
Reviewer 1:
- Line 244: Please change “and increased” to “but increased”.
Authors:
In accordance with the Reviewer’s recommendation, changes were added in the text of the Manuscript (line 293 in Mark-up Copy_R1).
Reviewer 1:
- Line 245: Please change “was observed more rarely” to “was more rarely observed.”.
Authors:
In accordance with the Reviewer’s recommendation, changes were added in the text of the Manuscript (line 295 in Mark-up Copy_R1).
Reviewer 1:
- Line 248: Please remove “in real time” after the acronym (RT–PCR) in: “Real-time reverse transcription polymerase chain reaction (RT–PCR) in real time”.
Authors:
In accordance with the Reviewer’s recommendation, changes were added in the text of the Manuscript (lines 298-299 in Mark-up Copy_R1).
Reviewer 1:
- Line 250: Please add spaces in between commas and letters in: “(Figure 3d,e,f) and FC (Figure 3g,h,k)”.
Authors:
In accordance with the Reviewer’s recommendation, changes were added in the text of the Manuscript (line 301 in Mark-up Copy_R1).
Reviewer 1:
- Line 254: Please add spaces in between commas and letters in: “(Figure 3d,e,f)”.
Authors:
In accordance with the Reviewer’s recommendation, changes were added in the text of the Manuscript (line 305 in Mark-up Copy_R1).
Reviewer 1:
- Line 255: Please add spaces in between commas and letters in: “(Figure 3g,h,k)”.
Authors:
In accordance with the Reviewer’s recommendation, changes were added in the text of the Manuscript (line 306 in Mark-up Copy_R1).
Reviewer 1:
- Line 258: Please simplify “obtained in comparisons between” to “obtained from comparing”.
Authors:
In accordance with the Reviewer’s recommendation, changes were added in the text of the Manuscript (lines 311-312 in Mark-up Copy_R1).
Reviewer 1:
- Line 259-260: Please simplify “obtained in comparisons between the” to “obtained from comparing the”.
Authors:
In accordance with the Reviewer’s recommendation, changes were added in the text of the Manuscript (line 312 in Mark-up Copy_R1).
Reviewer 1:
- Line 284: Please change “of sets pathways” to “of pathways sets”.
Authors:
In accordance with the Reviewer’s recommendation, changes were added in the text of the Manuscript (line 342 in Mark-up Copy_R1).
Reviewer 1:
- Line 287: Please correct “As a result, more than hundred pathways were” to “As a result, more than a hundred pathways were” or “As a result, more than one hundred pathways were”.
Authors:
In accordance with the Reviewer’s recommendation, changes were added in the text of the Manuscript (line 345 in Mark-up Copy_R1).
Reviewer 1:
- Line 295: Please correct “This pathway group we named by Pathway Cluster 1” to “This pathway group was denominated Pathway Cluster 1”.
Authors:
In accordance with the Reviewer’s recommendation, changes were added in the text of the Manuscript (line 354 in Mark-up Copy_R1).
Reviewer 1:
- Line 297: I kindly suggest changing “regardless FC or striatum” to “regardless being FC or striatum tissue”.
Authors:
In accordance with the Reviewer’s recommendation, changes were added in the text of the Manuscript (lines 356-357 in Mark-up Copy_R1).
Reviewer 1:
- Line 300: I kindly suggest changing “there were 10 and 7 pathways” to “there were identified 10 and 7 pathways”.
Authors:
In accordance with the Reviewer’s recommendation, changes were added in the text of the Manuscript (line 359 in Mark-up Copy_R1).
Reviewer 1:
- Line 301: I kindly suggest simplifying “They were named by” to “They were named”.
Authors:
In accordance with the Reviewer’s recommendation, changes were added in the text of the Manuscript (line 360 in Mark-up Copy_R1).
Reviewer 1:
- Lines 307: Please correct “Semax predominantly compensate of” to “Semax predominantly compensates for”.
Authors:
In accordance with the Reviewer’s recommendation, changes were added in the text of the Manuscript (line 366 in Mark-up Copy_R1).
Reviewer 1:
- Line 312: Please correct “genes were associated with NC” to “289 genes associated to NC”.
Authors:
We are grateful to the Reviewer for the comment. The changes were added in the text of the Manuscript (line 374 in Mark-up Copy_R1).
Reviewer 1:
- Line 314: Please correct “We calculated number of up- and (…)” to “We calculated a number of 289 of up- and (…).”.
Authors:
In accordance with the Reviewer’s recommendation, changes were added in the text of the Manuscript (line 314 in Mark-up Copy_R1).
Reviewer 1:
- Line 316: Please remove 289 gene from: “289 gene, Figure 2e, Supplementary Table S11)”.
Authors:
In accordance with the Reviewer’s recommendation, changes were added in the text of the Manuscript (line 376 in Mark-up Copy_R1).
Reviewer 1:
- Line 317: Please correct “as from overlapping genes” to “as 294 genes overlapping genes”.
Authors:
In accordance with the Reviewer’s recommendation, changes were added in the text of the Manuscript (line 378 in Mark-up Copy_R1).
Reviewer 1:
- Lines 317-318: Please remover 294 genes from: “(294 genes, Figure 2b)”.
Authors:
In accordance with the Reviewer’s recommendation, changes were added in the text of the Manuscript (lines 379 in Mark-up Copy_R1).
Reviewer 1:
- Line 318: Please correct “and co-directly” to “and 152 genes co-directly”.
Authors:
In accordance with the Reviewer’s recommendation, changes were added in the text of the Manuscript (line 379 in Mark-up Copy_R1).
Reviewer 1:
- Line 318: Please remove 152 genes from: “(152 genes, Figure 2b)”.
Authors:
In accordance with the Reviewer’s recommendation, changes were added in the text of the Manuscript (line 379 in Mark-up Copy_R1).
Reviewer 1:
- Line 336: Please correct “Top 5 the most significant” to “Top 5 of the most significant”.
Authors:
In accordance with the Reviewer’s recommendation, changes were added in the text of the Manuscript (line 398 in Mark-up Copy_R1).
Reviewer 1:
- Line 363: Please correct “In the scheme, pathway clusters (NC and IC) are grouped in the white” to “The scheme shows pathway clusters (NC and IC) grouped in white”.
Authors:
In accordance with the Reviewer’s recommendation, changes were added in the text of the Manuscript (line 430 in Mark-up Copy_R1).
Reviewer 1:
- Line 369: Please correct “with pathway clusters” to “to pathway clusters”.
Authors:
In accordance with the Reviewer’s recommendation, changes were added in the text of the Manuscript (line 438 in Mark-up Copy_R1).
Reviewer 1:
- Line 390: Please correct “associated with effects of” to “associated to effects of”.
Authors:
In accordance with the Reviewer’s recommendation, changes were added in the text of the Manuscript (line 461 in Mark-up Copy_R1).
Reviewer 1:
- Line 393: Please correct “Only those genes were included” to “The only genes included were”.
Authors:
In accordance with the Reviewer’s recommendation, changes were added in the text of the Manuscript (lines 464-465 in Mark-up Copy_R1).
Reviewer 1:
- Line 394: Please remove the word that from: “network that were DEGs”.
Authors:
In accordance with the Reviewer’s recommendation, changes were added in the text of the Manuscript (line 465 in Mark-up Copy_R1).
Reviewer 1:
- Lines 407-408: Please correct “those of them that reduced expression” to “those which reduced expression”.
Authors:
In accordance with the Reviewer’s recommendation, changes were added in the text of the Manuscript (line 481 in Mark-up Copy_R1).
Reviewer 1:
- Line 408: Please correct “vice versa” to “vice-versa”.
Authors:
In accordance with the Reviewer’s recommendation, changes were added in the text of the Manuscript (line 482 in Mark-up Copy_R1).
Reviewer 1:
- Line 409: Please correct “those of them that increased expression” to “those which increased expression”.
Authors:
In accordance with the Reviewer’s recommendation, changes were added in the text of the Manuscript (line 482 in Mark-up Copy_R1).
Reviewer 1:
- Line 416: Please correct “than thousand genes” to “than one thousand genes”.
Authors:
In accordance with the Reviewer’s recommendation, changes were added in the text of the Manuscript (line 489 in Mark-up Copy_R1).
Reviewer 1:
- Line 416: Please correct “and specific to FC” to “and were specific to FC”.
Authors:
In accordance with the Reviewer’s recommendation, changes were added in the text of the Manuscript (line 490 in Mark-up Copy_R1).
Reviewer 1:
- Line 422: Please remove “that were” from: “the number of genes that were overlapping between”.
Authors:
In accordance with the Reviewer’s recommendation, changes were added in the text of the Manuscript (line 496 in Mark-up Copy_R1).
Reviewer 1:
- Line 435: Please correct “In the frontal cortex” to “In FC”.
Authors:
In accordance with the Reviewer’s recommendation, changes were added in the text of the Manuscript (line 510 in Mark-up Copy_R1).
Reviewer 1:
- Line 437: Please correct “(…) ischemia, activation of genes involved (…)” to “ischemia, it was observed activation of genes involved (…)”.
Authors:
We are grateful to the Reviewer for the comment. The changes were added in the text of the Manuscript (lines 511-515 in Mark-up Copy_R1).
Reviewer 1:
- Line 438: Please remove “were observed” from “with neurotransmission were observed”.
Authors:
We are grateful to the Reviewer for the comment. The changes were added in the text of the Manuscript (lines 511-515 in Mark-up Copy_R1).
Reviewer 1:
- Line 441: Please include a comma in: “Using both RNA (…)” as in “Using both, RNA (…)”.
Authors:
In accordance with the Reviewer’s recommendation, changes were added in the text of the Manuscript (line 518 in Mark-up Copy_R1).
Reviewer 1:
- Line 441: Please include a comma in “real-time PCR on an (…)”, as in “real-time PCR, on an (…)”.
Authors:
In accordance with the Reviewer’s recommendation, changes were added in the text of the Manuscript (line 518 in Mark-up Copy_R1).
Reviewer 1:
- Line 453: Please remove “by us” in “observed by us in the present study (…)”.
Authors:
In accordance with the Reviewer’s recommendation, changes were added in the text of the Manuscript (line 532 in Mark-up Copy_R1).
Reviewer 1:
- Line 456: Please correct: “Such a result we also observed earlier both” to “Such a result we was also observed earlier, at both (…)”.
Authors:
We are grateful to the Reviewer for the comment. The changes were added in the text of the Manuscript (line 535 in Mark-up Copy_R1).
Reviewer 1:
- Line 462: Please correct “which we observed 24 hours after tMCAO (…)” to “which we was observed 24 hours after tMCAO (…)”.
Authors:
We are grateful to the Reviewer for the comment. The changes were added in the text of the Manuscript (line 543 in Mark-up Copy_R1).
Reviewer 1:
- Line 472: Please insert a comma in “generated both common (…)” as follows “generated both, common (…)”.
Authors:
In accordance with the Reviewer’s recommendation, changes were added in the text of the Manuscript (line 553 in Mark-up Copy_R1).
Reviewer 1:
- Line 477: Please correct “the effect of peptides manifests itself differently (…)” to “the effect of peptides its effect manifests itself differently (…)”.
Authors:
In accordance with the Reviewer’s recommendation, changes were added in the text of the Manuscript (lines 558-559 in Mark-up Copy_R1).
Reviewer 1:
- Line 478: Please correct “can be used in studying the targeted effect of drugs (…)” to “can be used in studying the targeted effect of for target drugs (…)”.
Authors:
In accordance with the Reviewer’s recommendation, changes were added in the text of the Manuscript (lines 559-560in Mark-up Copy_R1).
Reviewer 1:
- Line 482: Please correct “(…) associated with both peptides impact (..)” to “predominantly associated with both, peptides impact(..)”.
Authors:
In accordance with the Reviewer’s recommendation, changes were added in the text of the Manuscript (lines 564-565 in Mark-up Copy_R1).
Reviewer 1:
- Line 482: Please remove predominantly from the end of the sentence: “(…) nerve impulse transmission, predominantly.”.
Authors:
In accordance with the Reviewer’s recommendation, changes were added in the text of the Manuscript (lines 564-565 in Mark-up Copy_R1).
Reviewer 1:
- Line 487: Please correct “Ischemia had predominantly the opposite effect (…)” to “Ischemia had mostly the opposite effect (…)”.
Authors:
In accordance with the Reviewer’s recommendation, changes were added in the text of the Manuscript (line 569 in Mark-up Copy_R1).
Reviewer 1:
- Line 490: Please correct “We then suggested that there is (…)” to “It suggested that there is (…)”.
Authors:
In accordance with the Reviewer’s recommendation, changes were added in the text of the Manuscript (line 537 in Mark-up Copy_R1).
Reviewer 1:
- Line 493: I kindly suggest correcting “which is predominantly reduced by the peptide (…)” to “which is mainly reduced by the peptide (…)”.
Authors:
In accordance with the Reviewer’s recommendation, changes were added in the text of the Manuscript (line 576 in Mark-up Copy_R1).
Reviewer 1:
- Line 494: I kindly suggest correcting “the expression of which can be enhanced (…) to “whose expression can be enhanced (…)”.
Authors:
In accordance with the Reviewer’s recommendation, changes were added in the text of the Manuscript (line 577 in Mark-up Copy_R1).
Reviewer 1:
- Lines 495-496: Please correct “Here, we observed a group of IC genes, (…)” to “Here, it was seen a group of IC genes, (…)”.
Authors:
In accordance with the Reviewer’s recommendation, changes were added in the text of the Manuscript (line 579 in Mark-up Copy_R1).
Reviewer 1:
- Line 499: Please correct “and their receptors (…)” to “and respective receptors (…)”.
Authors:
In accordance with the Reviewer’s recommendation, changes were added in the text of the Manuscript (line 582 in Mark-up Copy_R1).
Reviewer 1:
- Line 500: I kindly suggest using “was IL-10 receptor” instead of “was interleukin 10 receptor (…)”.
Authors:
In accordance with the Reviewer’s recommendation, changes were added in the text of the Manuscript (line 583 in Mark-up Copy_R1).
Reviewer 1:
- Lines 503-505: I kindly suggest using “Furthermore, and in accordance with literature data, an increase in the expression of genes such as Tlr2, Tlr4, Olr1 (First gene group, Figure 5) points to the absence of a neuroprotective effect of the peptide in the striatum [39–41].” instead of “Furthermore, an increase in the expression of genes such as Tlr2, Tlr4, Olr1 (First gene group, Figure 5) indicated the absence of a neuroprotective effect of the peptide in the striatum, in accordance with literature data. [39–41].”.
Authors:
In accordance with the Reviewer’s recommendation, changes were added in the text of the Manuscript (lines 586-591 in Mark-up Copy_R1).
Reviewer 1:
- Line 508: Please correct “also modulated expression of NC genes (…)” to “also modulated the expression of NC genes (…)”.
Authors:
In accordance with the Reviewer’s recommendation, changes were added in the text of the Manuscript (line 594 in Mark-up Copy_R1).
Reviewer 1:
- Line 550: Please correct “can explain the differences we observed in action of (…)” to “can explain the differences observed in action of (…)”.
Authors:
In accordance with the Reviewer’s recommendation, changes were added in the text of the Manuscript (line 641 in Mark-up Copy_R1).
Reviewer 1:
- Line 565: I kindly suggest correcting “from the lesion and taking into account (…)” to “from the lesion while taking into account (…)”.
Authors:
In accordance with the Reviewer’s recommendation, changes were added in the text of the Manuscript (line 657 in Mark-up Copy_R1).
Reviewer 1:
- Lines 571-575: I kindly suggest paraphrasing the paragraph “Furthermore, further protein studies very significant to translate genome effects to phenotyping and clinical level. We believe that genome and metabolic responses identified not only local but entire brain can be the key for induction of regeneration processes in brain cells after stroke.” My humble suggestion for a more understandable reading:
“Further protein studies would be significant for translating genome effects into phenotype meaning and into clinical application as well. We believe that identifying genome and metabolic responses, not only locally but also considering the entire brain tissue, can be key for successfully inducting the regeneration processes in brain cells after stroke.”.
Authors:
In accordance with the Reviewer’s recommendation, changes were added in the text of the Manuscript (lines 663-671 in Mark-up Copy_R1).
Reviewer 1:
- Line 586: "Koizumi et al. ". Please correct to "Koizumi et al.,”.
Authors:
In accordance with the Reviewer’s recommendation, changes were added in the text of the Manuscript (line 685 in Mark-up Copy_R1).
Reviewer 1:
- Line 594: “respectively, were formed.”. Please correct to: “were respectively formed.”.
Authors:
In accordance with the Reviewer’s recommendation, changes were added in the text of the Manuscript (lines 699-700 in Mark-up Copy_R1).
Reviewer 1:
- Line 601: “Then total RNA (…)”. Please correct to: “Then, total RNA (…)”.
Authors:
We are grateful to the Reviewer for the comment. The changes were added in the text of the Manuscript (lines 713-716 in Mark-up Copy_R1).
Reviewer 1:
- Line 620: “as previously described”. To avoid repetition, kindly suggest: “as described in the past”.
Authors:
In accordance with the Reviewer’s recommendation, changes were added in the text of the Manuscript (line 727 in Mark-up Copy_R1).
Reviewer 1:
- Line 650: Please correct “genes that associated with action of ACTH-like peptides (…)” to “genes associated to the action of ACTH-like peptides (…)”.
Authors:
In accordance with the Reviewer’s recommendation, changes were added in the text of the Manuscript (line 786 in Mark-up Copy_R1).
Reviewer 1:
- Line 656: Please correct “compensatory effect on the (…)” to “compensatory effect upon the (…)”.
Authors:
In accordance with the Reviewer’s recommendation, changes were added in the text of the Manuscript (line 793 in Mark-up Copy_R1).
Reviewer 1:
- Lines 660-661: Please correct “peptide exhibit a heterogeneous effect (…)” to “peptide exhibited a heterogeneous effect (…)”.
Authors:
In accordance with the Reviewer’s recommendation, changes were added in the text of the Manuscript (line 798 in Mark-up Copy_R1).
Reviewer 1:
- Lines 661-663: I suggest paraphrasing the sentence “Thus, we revealed genes, which are subject to the greatest variability in changes in expression in different parts of the damaged hemisphere under the influence of the neuroprotective peptides.” I humbly leave a suggestion:
“And so, our study identified genes which are subject to a highly variable expression within different parts of the damaged hemisphere under the influence of the neuroprotective peptides.
Authors:
In accordance with the Reviewer’s recommendation, changes were added in the text of the Manuscript (lines 799-803 in Mark-up Copy_R1).
Reviewer 1:
- Line 663-665: I kindly suggest paraphrasing the sentence “We believe that transcriptome responses of such genes identified can be the key for induction of regeneration processes in brain cells after stroke.” to “We believe that the transcriptome responses of the identified genes can be the key for efficiently inducting regeneration processes in brain cells after stroke.”.
Authors:
In accordance with the Reviewer’s recommendation, changes were added in the text of the Manuscript (lines 803-806 in Mark-up Copy_R1).
Reviewer 1:
- Line 666: Please correct “tissues- and damage-specific” to “specific-tissue/damage”.
Authors:
In accordance with the Reviewer’s recommendation, changes were added in the text of the Manuscript (line 808 in Mark-up Copy_R1).

Reviewer 2 Report
Comments and Suggestions for Authors
This manuscript describes the results of bulk RNA sequencing assessment of the effects of two ACTH-related peptides on gene expression in the striatum of rats with ischemia-reperfusion injury by transient middle cerebral artery occlusion. The analysis pipeline is reasonable, and the results are interesting. The English usage makes some aspects of the manuscript difficult to understand, primarily regarding some unusual wordings, although most of the prose is easy to understand. The manuscript would benefit from a few clarifications as well.
- Introduction: It is not clear from the introduction why the ACTH-related peptides Semax and ACTH (6-9) were investigated in this experiment, aside from the fact that they had been looked at by other groups in the past. The manuscript would benefit greatly from a little more in the introduction (e.g. what is their proposed signaling pathway, or the rationale for why they were developed); more detail should be given in the discussion, to put these results in their place in the literature.
- A number of acronyms in the manuscript are not defined until the methods section at the end of the manuscript. They should be defined on first use (e.g. IA-s, -f, etc).
- Line 119 – states that peptides were given prior to tMCAO, but from the methods it seems that the peptide treatment was given after. Please keep this consistent with whichever time the peptide treatments were given.
- Figure 3 d-k: There are asterisks for statistically significant differences on these graphs. However, it is unclear what is being compared to what, and the statistical test used is not given. Please clarify.
- Line 310-311: states that ACTH(6-9)-PGP “recapitulated the effects of ischemia…” This statement is not justified by the data because there is no ACTH(6-9)-PGP only group (ie the effects of this peptide treatment alone are not tested). Thus, we don’t know whether this peptide would recapitulate the effects of ischemia.
- For Fig. 5, it is difficult to tell what genes point to each gene cluster in the network diagrams. It would be helpful to have a table that shows all the DEGs that belong to ach of these clusters shown. Alternatively, the clusters could be color-coded and the lines pointing to each also coded the same color to make it more obvious.
Comments on the Quality of English LanguageThere is some terminology used that is not standard English and it is difficult to understand the results because of this. The most prominent is co-directly and opposite-directly, which seem to mean that changes are in the same direction or opposite directions. This needs to be clarified.
Author Response
Response to the comments of Reviewer 2 to Manuscript ID: ijms-3685289
Authors:
We are very grateful to the Reviewer 2 for the review and constructive comments. We carefully considered the comments of the Reviewer 2 and attached the answers to all comments.
Reviewer 2:
- Introduction: It is not clear from the introduction why the ACTH-related peptides Semax and ACTH (6-9) were investigated in this experiment, aside from the fact that they had been looked at by other groups in the past. The manuscript would benefit greatly from a little more in the introduction (e.g. what is their proposed signaling pathway, or the rationale for why they were developed); more detail should be given in the discussion, to put these results in their place in the literature.
Authors:
In accordance with the Reviewer’s recommendation, changes were added in the text of the Manuscript (lines 70-85, 637-641 in Mark-up Copy_R1).
Reviewer 2:
- A number of acronyms in the manuscript are not defined until the methods section at the end of the manuscript. They should be defined on first use (e.g. IA-s, -f, etc).
Authors:
In accordance with the Reviewer’s recommendation, changes were added in the text of the Manuscript (lines 149-150, 160, 161, 183, 289, 290 in Mark-up Copy_R1).
Reviewer 2:
- Line 119 – states that peptides were given prior to tMCAO, but from the methods it seems that the peptide treatment was given after. Please keep this consistent with whichever time the peptide treatments were given.
Authors:
The rats received intraperitoneal administration of peptide (Semax or ACTH(6-9)PGP) at 1.5 h, 2.5 h and 6.5 h after the beginning of the surgical operation (tMCAO).
In accordance with the Reviewer’s recommendation, changes were added in the text of the Manuscript (lines 159, 276, 697 in Mark-up Copy_R1).
Reviewer 2:
- Figure 3 d-k: There are asterisks for statistically significant differences on these graphs. However, it is unclear what is being compared to what, and the statistical test used is not given. Please clarify.
Authors:
The comparison was within IR-f vs. SO-f, IS-f vs. IR-f, IA-f vs. IR-f, IR-s vs. SO-s, IS-s vs. IR-s, and IA-s vs. IR-s pairwise comparisons. PCR calculations were performed using the Relative Expression Software Tool (REST) 2005 software. When comparing groups of PCR data, differences with a p-value less than 0.05 (two-sided Pair-Wise Fixed Reallocation Randomization Test) were considered as a significant. RNA-Seq results present for comparison. Padj is the p-value (t-test) adjusted with the Benjamini-Hochberg correction. Only genes with Padj below 0.05 (Padj < 0.05) and >1.5-fold changes in expression were taken as significant.
In accordance with the Reviewer’s recommendation, changes were added in the Figure 3 and the text of the Manuscript (lines 318-322 in Mark-up Copy_R1).
Reviewer 2:
- Line 310-311: states that ACTH(6-9)-PGP “recapitulated the effects of ischemia…” This statement is not justified by the data because there is no ACTH(6-9)-PGP only group (ie the effects of this peptide treatment alone are not tested). Thus, we don’t know whether this peptide would recapitulate the effects of ischemia.
Authors:
In accordance with the Reviewer’s recommendation, changes were added in the text of the Manuscript (line 370 in Mark-up Copy_R1).
Reviewer 2:
- For Fig. 5, it is difficult to tell what genes point to each gene cluster in the network diagrams. It would be helpful to have a table that shows all the DEGs that belong to ach of these clusters shown. Alternatively, the clusters could be color-coded and the lines pointing to each also coded the same color to make it more obvious.
Authors:
In accordance with the Reviewer’s recommendation, Supplementary Table S13 to show DEGs that belong to the different clusters was added. Furthermore, changes were added in the text of the Manuscript (lines 425, 440, 456, 831-832 in Mark-up Copy_R1).
Reviewer 2:
Comments on the Quality of English Language: There is some terminology used that is not standard English and it is difficult to understand the results because of this. The most prominent is co-directly and opposite-directly, which seem to mean that changes are in the same direction or opposite directions. This needs to be clarified.
Authors:
In accordance with the Reviewer’s recommendation, the changes were added in the text (lines 174, 219-220 in Mark-up Copy_R1). Furthermore, the text of the manuscript was undergone by overall revision of the English language. The changes were added throughout the text (file “Mark-up Copy_R1”).

Round 2
Reviewer 1 Report
Comments and Suggestions for Authors
The manuscript presentation has clearly improved with the changes. Congratulations to the authors. Please. make sure references are in accord with Journal's rules.
Comments on the Quality of English LanguageRecommended proofreading to make sure of an English language high quality.